# Error dynamics of mini-batch gradient descent with random reshuffling for least squares regression

**Jackie Lok**                                                    JACKIE.LOK@PRINCETON.EDU
*ORFE Department, Princeton University*

**Rishi Sonthalia**                                              RISHI.SONTHALIA@BC.EDU
*Department of Mathematics, Boston College*

**Elizaveta Rebrova**                                            ELRE@PRINCETON.EDU
*ORFE Department, Princeton University*

Editors: Gautam Kamath and Po-Ling Loh

## Abstract

We study the discrete dynamics of mini-batch gradient descent with random reshuffling for least squares regression. We show that the training and generalization errors depend on a sample cross-covariance matrix $\mathbf{Z}$ between the original features $\mathbf{X}$ and a set of new features $\widetilde{\mathbf{X}}$ in which each feature is modified by the mini-batches that appear before it during the learning process in an averaged way. Using this representation, we establish that the dynamics of mini-batch and full-batch gradient descent agree up to leading order with respect to the step size using the linear scaling rule. However, mini-batch gradient descent with random reshuffling exhibits a subtle dependence on the step size that a gradient flow analysis cannot detect, such as converging to a limit that depends on the step size. By comparing $\mathbf{Z}$, a non-commutative polynomial of random matrices, with the sample covariance matrix of $\mathbf{X}$ asymptotically, we demonstrate that batching affects the dynamics by resulting in a form of shrinkage on the spectrum.

## 1. Introduction

Modern machine learning models are primarily trained via gradient based methods on large datasets. Since it is typically not feasible to compute the entire gradient, stochastic gradient descent (SGD) and its variants are often the algorithm of choice (Bottou, 2012). In a variant of SGD known as *mini-batch gradient descent*, a subset of the training data, or *mini-batch*, is used in each iteration. Studying the dynamics of gradient descent is an important problem for understanding the training dynamics and generalization capabilities of the learned parameters, especially for overparameterized models (Gunasekar et al., 2018a,b). However, the effects of mini-batching on the error dynamics are less well-understood.

There are also different ways to sample the mini-batch in each iteration. The most commonly studied method is sampling with replacement, where a random subset of data is used to select the mini-batch in each iteration. Thus, in each epoch, the same data point may be used more than once. However, in practice, *random reshuffling* is typically used: at the beginning of each epoch, the dataset is partitioned into mini-batches, randomly permuted, and iterated through. It has been observed that sampling without replacement in this way often leads to faster convergence (Bottou, 2009, 2012). However, the introduction of dependencies between batches makes theoretical analysis of the dynamics more difficult (Gürbüzbalaban et al., 2021; HaoChen and Sra, 2019).

In this paper, we aim to contribute towards a better understanding of sampling without replacement. By analyzing the discrete dynamics of mini-batch gradient descent with random reshuffling

for the fundamental problem of least squares regression, we find that there are higher-order effects introduced by sampling without replacement that are not present when sampling with replacement, which result in subtly different trajectories.

**Contributions.** Our main contributions are the following:

- **Exact characterization of error dynamics.** We show that the training dynamics (Theorem 2) and generalization error (Theorem 8) of the mean iterate of mini-batch gradient descent with random reshuffling, averaged over the permutations of mini-batches in each epoch, are governed by a sample cross-covariance matrix $\mathbf{Z} := \frac{1}{n}\widetilde{\mathbf{X}}^{\mathsf{T}}\mathbf{X}$ that captures the interaction between the original features $\mathbf{X}$ and a set of modified features $\widetilde{\mathbf{X}}$ (defined in Section 3). The matrix $\mathbf{Z}$ encapsulates the influence of preceding mini-batches on each feature in an averaged manner, providing a framework for analyzing the learning process. Our results are stated under minimal assumptions on the data, learning rate, and mini-batches.

- **Comparison with full-batch gradient descent/sampling with replacement.** Our analysis demonstrates that the error dynamics of mini-batch gradient descent with random reshuffling are controlled by the sample cross-covariance matrix $\mathbf{Z}$ in a way that is analogous to how full-batch gradient descent (or SGD when sampling with replacement) depends on the sample covariance matrix $\mathbf{W} := \frac{1}{n}\mathbf{X}^{\mathsf{T}}\mathbf{X}$. We find that $\mathbf{Z}$, which is a non-commutative polynomial in the sample covariance matrices of each mini-batch, matches $\mathbf{W}$ up to leading order with respect to the step size $\alpha$. Based on this connection, we establish that the linear scaling rule, which calls for the step size to be scaled proportionally by the number of batches, matches the error dynamics of full-batch and mini-batch gradient descent for infinitesimal step sizes (Remark 5). However, for finite step sizes, mini-batch gradient descent with random reshuffling exhibits a subtle dependence on the step size that a continuous-time gradient flow analysis cannot detect; for example, it may converge to a step size-dependent limit that differs from the usual shifted minimum-norm solution obtained with full-batch gradient descent or SGD when sampling with replacement (Corollary 4).

- **Effects of batching.** We analyze the effects of batching on the error dynamics compared to full-batch gradient descent by comparing the asymptotic properties of the matrices $\mathbf{Z}$ and $\mathbf{W}$. As the number of data samples $n$ tends to infinity and the dimension of the parameters $p$ is fixed, we establish that asymptotically, while $\mathbf{Z}$ and $\mathbf{W}$ share the same eigenvectors, the eigenvalues of $\mathbf{Z}$ are systematically shrunk compared to those of $\mathbf{W}$ (Proposition 12), which directly affects the training and generalization errors (Proposition 13). Furthermore, we demonstrate that batching results in a similar effect in the more complicated proportional regime where $p/n \to \gamma \in (0, \infty)$ by numerically computing the limiting spectrum of $\mathbf{Z}$ under a more specific Gaussian random matrix model using tools from free probability theory (Section 3.3.2).

## 1.1. Related works

**Gradient flow.** The error dynamics of gradient descent has typically been analyzed from the perspective of continuous-time gradient flow, which is a good approximation assuming that infinitesimal learning rates are used. This perspective is adopted in Skouras et al. (1994); Advani et al. (2020); Ali et al. (2019, 2020) to study the effects of early stopping and implicit regularization via connections with ridge regression. One of the key themes in these works is that the trajectory and generalization error of (full-batch, continuous-time) gradient descent for least squares regression

are determined by the spectrum of the sample covariance matrix $\mathbf{W} = \frac{1}{n}\mathbf{X}^\mathsf{T}\mathbf{X}$ of the data. In our work, we consider the discrete dynamics of gradient descent with finite learning rates, and show that the error dynamics of mini-batch gradient descent with random reshuffling depends analogously on a cross-covariance matrix $\mathbf{Z} = \frac{1}{n}\widetilde{\mathbf{X}}^\mathsf{T}\mathbf{X}$ that involves a set of modified features $\widetilde{\mathbf{X}}$.

**SGD: Sampling with replacement.** Stochastic gradient descent has most commonly been studied assuming that the mini-batches are independently sampled with replacement in each iteration, which makes the process more amenable to theoretical analysis. From an optimization perspective, the convergence rates of SGD have been well-studied under various assumptions on the objective function and with different sampling schemes (Bach and Moulines, 2011, 2013; Needell et al., 2016; Ma et al., 2018; Gower et al., 2019). Explanations of the good generalization properties of SGD, based on properties such as the width of the final minima obtained or the optimal batch size and learning rate, have also been offered based on analogies with stochastic differential equations (Mandt et al., 2017; Smith and Le, 2018; Jastrzębski et al., 2018; Li et al., 2017, 2019a, 2021; Malladi et al., 2022). These works assume that the mini-batches are sampled independently with replacement in each epoch, and also typically assume that vanishing learning rates are used.

**SGD: Sampling without replacement.** A line of work that analyzes the implicit bias of SGD with finite learning rates uses the technique of backward error analysis, beginning with the analysis of gradient descent in Barrett and Dherin (2021); Miyagawa (2022), and extended to analyze SGD with random reshuffling—which is the same model that we consider—in Smith et al. (2021). Specifically, it is shown that the mean iterate, averaged over the permutations of mini-batches in each epoch, is close to the path of gradient flow on a modified loss with an additional regularization term that penalizes the norms of the mini-batch gradients. The mean evolution of SGD using sampling without replacement is also studied in Beneventano (2023) under weaker assumptions. Backward error analysis has also been used to show that adaptive algorithms such as Adam and RMSProp have similar implicit regularization in Cattaneo et al. (2024). Compared to these works, we consider linear models specifically instead of general loss functions; however, our results are presented with minimal assumptions and essentially apply to any input data, choice of mini-batches, batch size, and step size.

Another notable line of work from the stochastic optimization literature studies the convergence rates of SGD when sampling without replacement. In one of the earliest theoretical results, Gürbüzbalaban et al. (2021) shows that for quadratic objective functions (or more generally, strongly convex smooth objectives), SGD with random reshuffling, using a prescribed sequence of step sizes, converges asymptotically at a rate of $O(1/k^2)$ where $k$ is the number of epochs, which is superior to the $O(1/k)$ rate of SGD when sampling with replacement. In subsequent works (Shamir, 2016; HaoChen and Sra, 2019; Nagaraj et al., 2019; Rajput et al., 2020; Nguyen et al., 2021; Mishchenko et al., 2020), the complexity advantage of random reshuffling over sampling with replacement (with a primary focus on mini-batches of size one) is analyzed for more general optimization problems where assumptions such as strong convexity, smoothness, and bounded gradients are relaxed. In our work, we study random reshuffling from a different perspective for the special case of least squares regression by providing an exact description of the error dynamics for various batch sizes in terms of the spectrum of the data covariance matrix instead of complexity bounds. As such, our results are not directly comparable with these prior results.

**Linear scaling rule.** An important aspect of SGD is the choice of batch size and the learning rate. It has been observed that training with larger mini-batches can be more efficient (Smith et al., 2018; Geiping et al., 2022) and lead to better generalization properties (Li et al., 2019b; Lewkowycz et al., 2020). A connection between the batch size and the learning rate for SGD known as the *linear scaling rule* states that by adjusting the mini-batch size and learning rate proportionally by the same factor, the training dynamics do not change. This was empirically discovered to be a practically useful heuristic for training deep neural networks using SGD (Krizhevsky, 2014; Goyal et al., 2018; Smith et al., 2018; He et al., 2019), and theoretical explanations have been proposed based on the effect of noise on the estimation of the gradient in each mini-batch for SGD.[1] For more general convex losses, it is also shown in Ma et al. (2018) that SGD with mini-batch sizes below a certain threshold is consistent with the linear scaling rule. In our work, we use a different approach to find that the linear scaling rule emerges naturally from analyzing the dynamics of the mean SGD iterate for linear models under no assumptions on the batch size or the noise from mini-batch gradient estimation. Our approach also shows that the linear scaling rule can fail to hold (dramatically) for large step sizes; see Remark 6.

**Linear models.** Linear models in the high-dimensional regime have recently been extensively studied in the high-dimensional statistics literature, offering explanations for many interesting empirical phenomena in deep learning, such as double descent and the benefits of overparameterization. It has been shown that neural networks in a "lazy" training regime in which the weights do not change much around initialization are essentially equivalent to linear models (Chizat et al., 2019; Du et al., 2019b,a; Misiakiewicz and Montanari, 2023). The generalization errors of ridge(less) regression with random data are precisely described in Dobriban and Wager (2018); Hastie et al. (2022); Mei and Montanari (2022); Kausik et al. (2024). From a dynamical perspective, Paquette et al. (2021); Lee et al. (2022); Paquette et al. (2022) show that the trajectories of SGD for ridge regression with finite step sizes and high-dimensional random data concentrate on a deterministic function determined by a Volterra equation, assuming the batch sizes are vanishingly small as a fraction of the sample size. Analogous concentration results for the trajectories of SGD for a wider class of models such as two-layer neural networks have also been derived concurrently (Saad and Solla, 1995; Goldt et al., 2019; Ben Arous et al., 2022; Arnaboldi et al., 2023). Our work provides exact formulas for the training and generalization errors for linear models trained by mini-batch gradient descent with random reshuffling, establishing an analogy with the more well-studied dynamics of full-batch gradient descent or SGD with replacement.

## 2. Model

Suppose that we are given $n$ i.i.d. data samples $(\mathbf{x}_i, y_i)$, where $\mathbf{x}_i \in \mathbb{R}^p$ is the feature vector and $y_i \in \mathbb{R}$ is the response given by $y_i = \mathbf{x}_i^\mathsf{T} \boldsymbol{\beta}_* + \eta_i$, with $\boldsymbol{\beta}_* \in \mathbb{R}^p$ an underlying parameter vector and $\eta_i$ a noise term. We will assume that the (uncentered) covariance matrix of the features $\mathbf{x}_i$ is given by $\mathbb{E}\left[\mathbf{x}_i \mathbf{x}_i^\mathsf{T}\right] = \Sigma$, and the noise terms $\eta_i$ have mean $\mathbb{E}\left[\eta_i \mid \mathbf{x}_i\right] = 0$ and variance $\mathbb{E}\left[\eta_i^2 \mid \mathbf{x}_i\right] = \sigma^2$, conditional on the features. By arranging each observation as a row, we can write the linear model in matrix form as $\mathbf{y} = \mathbf{X}\boldsymbol{\beta}_* + \boldsymbol{\eta}$, where $\mathbf{y} \in \mathbb{R}^n$ and $\mathbf{X} \in \mathbb{R}^{n \times p}$.

---

1. Interestingly, different optimizers may have different scaling rules: a square root scaling rule has been derived for adaptive gradient algorithms such as Adam and RMSProp using random matrix theory (Granziol et al., 2022) and SDE approximation (Malladi et al., 2022).

We consider the following model of *mini-batch gradient descent with random reshuffling* using $B \geq 1$ mini-batches, initialized at $\boldsymbol{\beta}_0 \in \mathbb{R}^p$. For simplicity, we will assume that $B$ divides $n$. Suppose that the data $\mathbf{X}$ is partitioned into $B$ equally-sized mini-batches $\mathbf{X}_1, \ldots, \mathbf{X}_B \in \mathbb{R}^{(n/B) \times p}$, and let $\mathbf{y}_1, \ldots, \mathbf{y}_B$ and $\boldsymbol{\eta}_1, \ldots, \boldsymbol{\eta}_B$ denote the corresponding entries of $\mathbf{y}$ and $\boldsymbol{\eta}$. In each epoch, a permutation $\tau = (\tau(1), \tau(2), \ldots, \tau(B))$ of the $B$ mini-batches is chosen uniformly at random, and $B$ iterations of gradient descent with step size $\alpha$ are performed with respect to the loss functions

$$L_b(\boldsymbol{\beta}) := \frac{B}{2n} \|\mathbf{y}_b - \mathbf{X}_b \boldsymbol{\beta}\|_2^2 \tag{2.1}$$

for $b = \tau(1), \ldots, \tau(B)$ using this ordering. That is, if $\boldsymbol{\beta}_k^{(b)}$ denotes the parameters after the first $b$ iterations using the mini-batches $\mathbf{X}_{\tau(1)}, \ldots, \mathbf{X}_{\tau(b)}$ in the $k$th epoch, then

$$\boldsymbol{\beta}_k^{(b)} = \boldsymbol{\beta}_k^{(b-1)} - \frac{B\alpha}{n} \mathbf{X}_{\tau(b)}^{\mathsf{T}} (\mathbf{X}_{\tau(b)} \boldsymbol{\beta}_k^{(b-1)} - \mathbf{y}_{\tau(b)}), \quad b = 1, 2, \ldots, B, \tag{2.2}$$

with $\boldsymbol{\beta}_k^{(0)} := \boldsymbol{\beta}_{k-1}^{(B)}$ and $\boldsymbol{\beta}_0^{(B)} := \boldsymbol{\beta}_0$. Denote the set of all permutations of $B$ elements by $S_B$. Let

$$\bar{\boldsymbol{\beta}}_k := \mathbb{E}_{\tau \sim \mathrm{Unif}(S_B)} \left[ \boldsymbol{\beta}_k^{(B)} \right] \tag{2.3}$$

be the *mean iterate* after $k$ epochs, averaged over the random permutations of the mini-batches in each epoch. Note that *full-batch gradient descent* corresponds to $B = 1$ with this setup.

Our goal is to study the dynamics of the error vector $\bar{\boldsymbol{\beta}}_k - \boldsymbol{\beta}_*$ (i.e. the *training dynamics*), as well as the corresponding *generalization error* $R_{\mathbf{X}}(\bar{\boldsymbol{\beta}}_k)$, representing the prediction error on an out-of-sample observation, defined by

$$R_{\mathbf{X}}(\boldsymbol{\beta}) := \mathbb{E}_{\mathbf{x}, \boldsymbol{\eta}} \left[ (\mathbf{x}^{\mathsf{T}} \boldsymbol{\beta} - \mathbf{x}^{\mathsf{T}} \boldsymbol{\beta}_*)^2 \mid \mathbf{X} \right] = \mathbb{E}_{\boldsymbol{\eta}} \left[ \|\boldsymbol{\beta} - \boldsymbol{\beta}_*\|_{\Sigma}^2 \mid \mathbf{X} \right]. \tag{2.4}$$

Here, the expectation, conditional on the data $\mathbf{X}$, is taken over a newly sampled feature vector $\mathbf{x}$ and the randomness in $\boldsymbol{\eta}$, and $\|\mathbf{z}\|_{\Sigma}^2 = \mathbf{z}^{\mathsf{T}} \Sigma \mathbf{z}$ denotes the norm induced by $\Sigma$.

**Outline.** The rest of the paper is structured as follows. Section 3 describes our main results on analyzing mini-batch gradient descent. After defining the modified features $\widetilde{\mathbf{X}}$ and the matrix $\mathbf{Z}$, we provide exact formulae for the training dynamics and generalization error in Sections 3.1 and 3.2 respectively. In Section 3.3, we consider the asymptotic properties of $\mathbf{Z}$ to evaluate these expressions and provide more insights into the effects of batching. We will defer most of the proofs and technical details to the Appendix.

## 3. Analysis of mini-batch gradient descent with random reshuffling

In this section, we will show that the error dynamics of mini-batch gradient descent with random reshuffling are governed by a set of features that are modified by the other mini-batches. Specifically, for $b = 1, \ldots, B$, let $\mathbf{W}_b := \frac{B}{n} \mathbf{X}_b^{\mathsf{T}} \mathbf{X}_b$ denote the sample covariance matrix of each mini-batch, and define the *modified mini-batches* $\widetilde{\mathbf{X}}_b := \mathbf{X}_b \Pi_b$, where[2]

$$\Pi_b := \mathbb{E}_{\tau \sim \mathrm{Unif}(S_B)} \left[ \prod_{j: j < \tau^{-1}(b)} (\mathbf{I} - \alpha \mathbf{W}_{\tau(j)}) \right] = \frac{1}{B!} \sum_{\tau \in S_B} \prod_{j: j < \tau^{-1}(b)} (\mathbf{I} - \alpha \mathbf{W}_{\tau(j)}). \tag{3.1}$$

---

2. By convention, we identify each permutation $\tau$ in $S_B$, the set of all permutations of $B$ elements, with a list $(\tau(1), \tau(2), \ldots, \tau(B))$ of matrices that are multiplied from right to left in the product. Thus, $\tau^{-1}(b)$ denotes the position of mini-batch $b$ in the epoch. Furthermore, we take the product over an empty set to be the identity matrix.

That is, each feature $\mathbf{x}_i$ in $\mathbf{X}_b$ corresponds to the feature $\Pi_b \mathbf{x}_i$ in $\widetilde{\mathbf{X}}_b$, which has been modified by all the other mini-batches that appear before it in the learning process in an averaged way. Let $\widetilde{\mathbf{X}} \in \mathbb{R}^{n \times p}$ be the concatenation of the modified mini-batches $\widetilde{\mathbf{X}}_b$ in the same order as the original partition, and define

$$\mathbf{Z} := \frac{1}{n}\widetilde{\mathbf{X}}^\mathsf{T}\mathbf{X} = \frac{1}{n}\sum_{b=1}^{B} \Pi_b \mathbf{X}_b^\mathsf{T}\mathbf{X}_b \tag{3.2}$$

to be the $p \times p$ *sample cross-covariance matrix* of the modified features with the original features. The following technical lemma describes some key properties of $\mathbf{Z}$; its proof, which uses the properties of the symmetric group $S_B$ in the definition of $\widetilde{\mathbf{X}}_b$, can be found in Appendix B.1.1.

**Lemma 1** *Let $\widetilde{\mathbf{X}}$ and $\mathbf{Z}$ be defined as in* (3.1) *and* (3.2). *Then $\mathbf{Z}$ is a symmetric matrix, and hence all of its eigenvalues are real. Furthermore,* $\mathrm{Range}(\mathbf{Z}) \subseteq \mathrm{Range}(\widetilde{\mathbf{X}}^\mathsf{T}) \subseteq \mathrm{Range}(\mathbf{X}^\mathsf{T})$, *where* $\mathrm{Range}(\cdot)$ *denotes the column space of a matrix.*

Finally, observe that $\widetilde{\mathbf{X}} \equiv \widetilde{\mathbf{X}}(\alpha)$ and $\mathbf{Z} \equiv \mathbf{Z}(\alpha)$ are functions of the step size $\alpha$. In particular, it follows from the definition of the modified features $\widetilde{\mathbf{X}}_b = \mathbf{X}_b \Pi_b$ in (1) that we can write

$$\mathbf{Z}(\alpha) = \frac{1}{n}\sum_{b=1}^{B} \mathbf{X}_b^\mathsf{T}\mathbf{X}_b + O(\alpha) = \mathbf{W} + O(\alpha), \tag{3.3}$$

where $\mathbf{W} := \frac{1}{n}\mathbf{X}^\mathsf{T}\mathbf{X} = \frac{1}{n}\sum_{b=1}^{B}\mathbf{X}_b^\mathsf{T}\mathbf{X}_b$, and $O(\alpha)$ denotes terms of order $\alpha$ or smaller as $\alpha \to 0$. This shows that $\mathbf{Z}$ matches $\mathbf{W}$, the sample covariance matrix of the features, up to leading order in the step size $\alpha$. In general, $\mathbf{Z}$ *is a complicated non-commutative polynomial of the mini-batch sample covariance matrices* $\mathbf{W}_1, \ldots, \mathbf{W}_B$.

**Example 1 (Two-batch gradient descent)** *For a concrete example where we can write down a tractable, explicit expression for $\mathbf{Z}$, consider the case of* two-batch gradient descent *with $B = 2$ and mini-batches $\mathbf{X}_1, \mathbf{X}_2 \in \mathbb{R}^{(n/2) \times p}$. Here, the sample covariance matrices of the mini-batches are $\mathbf{W}_1 = \frac{2}{n}\mathbf{X}_1^\mathsf{T}\mathbf{X}_1$ and $\mathbf{W}_2 = \frac{2}{n}\mathbf{X}_2^\mathsf{T}\mathbf{X}_2$, and the modified mini-batches are given by*

$$\widetilde{\mathbf{X}}_1 \equiv \widetilde{\mathbf{X}}_1(\alpha) = \mathbf{X}_1\left(\mathbf{I} - \frac{1}{2}\alpha\mathbf{W}_2\right) \quad and \quad \widetilde{\mathbf{X}}_2 \equiv \widetilde{\mathbf{X}}_2(\alpha) = \mathbf{X}_2\left(\mathbf{I} - \frac{1}{2}\alpha\mathbf{W}_1\right). \tag{3.4}$$

*Thus, the features in $\widetilde{\mathbf{X}}_1$, corresponding to the first mini-batch, are given by $\left(\mathbf{I} - \frac{1}{2}\alpha\mathbf{W}_2\right)\mathbf{x}_i$. The sample cross-covariance matrix of the modified features $\widetilde{\mathbf{X}}$ with the original features is given by*

$$\mathbf{Z} \equiv \mathbf{Z}(\alpha) = \frac{1}{n}(\widetilde{\mathbf{X}}_1(\alpha)^\mathsf{T}\mathbf{X}_1 + \widetilde{\mathbf{X}}_2(\alpha)^\mathsf{T}\mathbf{X}_2) = \frac{1}{2}\left(\mathbf{I} - \frac{1}{2}\alpha\mathbf{W}_2\right)\mathbf{W}_1 + \frac{1}{2}\left(\mathbf{I} - \frac{1}{2}\alpha\mathbf{W}_1\right)\mathbf{W}_2$$

$$= \frac{1}{2}\left(\mathbf{W}_1 + \mathbf{W}_2\right) - \frac{1}{4}\alpha\left(\mathbf{W}_2\mathbf{W}_1 + \mathbf{W}_1\mathbf{W}_2\right). \tag{3.5}$$

*Since $\frac{1}{2}(\mathbf{W}_1 + \mathbf{W}_2) = \frac{1}{n}(\mathbf{X}_1^\mathsf{T}\mathbf{X}_1 + \mathbf{X}_2^\mathsf{T}\mathbf{X}_2) = \mathbf{W}$, it is easily seen that $\mathbf{Z} = \mathbf{W} + O(\alpha)$. Even in this simple setting, $\mathbf{Z}$ is already non-trivial to analyze since it involves interactions between the two mini-batches in the term $\mathbf{W}_2\mathbf{W}_1 + \mathbf{W}_1\mathbf{W}_2$, known as the* anticommutator *of $\mathbf{W}_1$ and $\mathbf{W}_2$.*

## 3.1. Training error dynamics

First, we derive an expression for the dynamics of the mean error $\bar{\boldsymbol{\beta}}_k - \boldsymbol{\beta}_*$ under mini-batch gradient descent with random reshuffling. The expression depends on *the spectrum of the sample cross-covariance matrix* $\mathbf{Z}$ and *the alignment of the initial error* $\boldsymbol{\beta}_0 - \boldsymbol{\beta}_*$ *with the eigenspaces of* $\mathbf{Z}$.

**Theorem 2** *Let* $\bar{\boldsymbol{\beta}}_k \in \mathbb{R}^p$ *be the mean iterate after* $k$ *epochs of gradient descent with* $B$ *minibatches, step size* $\alpha \geq 0$*, and initialization* $\boldsymbol{\beta}_0 \in \mathbb{R}^p$*. Let* $\widetilde{\mathbf{X}} \in \mathbb{R}^{n \times p}$ *be defined as in* (3.1) *and* $\mathbf{Z} = \frac{1}{n}\widetilde{\mathbf{X}}^\mathsf{T}\mathbf{X}$*, and assume that* $\mathrm{Range}(\widetilde{\mathbf{X}}^\mathsf{T}) \subseteq \mathrm{Range}(\widetilde{\mathbf{X}}^\mathsf{T}\mathbf{X})$*. Then for all* $k \geq 0$*,*

$$\bar{\boldsymbol{\beta}}_k - \boldsymbol{\beta}_* = (\mathbf{I} - B\alpha\mathbf{Z})^k(\boldsymbol{\beta}_0 - \boldsymbol{\beta}_*) + \frac{1}{n}\left[\mathbf{I} - (\mathbf{I} - B\alpha\mathbf{Z})^k\right]\mathbf{Z}^\dagger\widetilde{\mathbf{X}}^\mathsf{T}\boldsymbol{\eta}. \tag{3.6}$$

*Furthermore, if* $\mathbf{P}_{\mathbf{Z},0} := \mathbf{I} - \mathbf{Z}^\dagger\mathbf{Z}$ *and* $\mathbf{P}_\mathbf{Z} := \mathbf{I} - \mathbf{P}_{\mathbf{Z},0}$ *denote the orthogonal projectors onto the nullspace and row (or column) space of* $\mathbf{Z}$ *respectively (where* $(\cdot)^\dagger$ *is the Moore–Penrose pseudoinverse of a matrix), then we may decompose the first term as*

$$(\mathbf{I} - B\alpha\mathbf{Z})^k(\boldsymbol{\beta}_0 - \boldsymbol{\beta}_*) = \mathbf{P}_{\mathbf{Z},0}(\boldsymbol{\beta}_0 - \boldsymbol{\beta}_*) + (\mathbf{I} - B\alpha\mathbf{Z})^k\mathbf{P}_\mathbf{Z}(\boldsymbol{\beta}_0 - \boldsymbol{\beta}_*). \tag{3.7}$$

The proof of Theorem 2 is given in Appendix B.1.2; the main technical part involves developing some algebraic identities relating $\mathbf{Z}$ and products of the form $\mathbf{I} - \alpha\mathbf{W}_b$ for each mini-batch. The requirement $\mathrm{Range}(\widetilde{\mathbf{X}}^\mathsf{T}) \subseteq \mathrm{Range}(\widetilde{\mathbf{X}}^\mathsf{T}\mathbf{X})$ is purely a technical assumption to ensure that $\mathbf{P}_\mathbf{Z}\widetilde{\mathbf{X}}^\mathsf{T} = \widetilde{\mathbf{X}}^\mathsf{T}$ in order to control the learned noise, otherwise the iterate will always diverge.[3] The requirement appears to be generic; for example, in the overparameterized regime where $p \geq n$, it simply follows from the natural assumption that $\mathbf{X}$ has full rank.

The first term $\mathbf{P}_{\mathbf{Z},0}(\boldsymbol{\beta}_0 - \boldsymbol{\beta}_*)$ of (3.7) corresponds to the components of $\boldsymbol{\beta}_0 - \boldsymbol{\beta}_*$ that cannot be learned by mini-batch gradient descent with random reshuffling—referred to as a *"frozen subspace"* of weights in Advani et al. (2020) in the context of (full-batch) gradient descent—and the second term $\mathbf{P}_\mathbf{Z}(\boldsymbol{\beta}_0 - \boldsymbol{\beta}_*)$ corresponds to the *learnable components*. In particular, note that the projector $\mathbf{P}_{\mathbf{Z},0}$ is always non-trivial in the overparameterized regime where $p > n$.

### 3.1.1. COMPARISON WITH FULL-BATCH AND MINI-BATCHING WITH REPLACEMENT

First, we recall the known result that the full-batch gradient descent iterate $\widehat{\boldsymbol{\beta}}_k$ satisfies the following (for a proof and additional background, we refer to Appendix A):

$$\widehat{\boldsymbol{\beta}}_k - \boldsymbol{\beta}_* = (\mathbf{I} - \alpha\mathbf{W})^k(\boldsymbol{\beta}_0 - \boldsymbol{\beta}_*) + \frac{1}{n}\left[\mathbf{I} - (\mathbf{I} - \alpha\mathbf{W})^k\right]\mathbf{W}^\dagger\mathbf{X}^\mathsf{T}\boldsymbol{\eta}. \tag{3.8}$$

**Remark 3 (Sampling with replacement)** *Suppose that in each iteration, we sample a mini-batch with replacement uniformly at random from the fixed set of* $B$ *mini-batches* $\mathbf{X}_1, \ldots, \mathbf{X}_B$ *instead. Then it can be shown that after* $k$ *epochs (or* $Bk$ *iterations), the error corresponding to the mean iterate of this sampling process also satisfies* (3.8) *up to a time change by a factor of* $B$*; i.e. the same equation holds with* $k$ *replaced by* $Bk$*. For the details, see Appendix B.1.4.*

Therefore, comparing (3.8) with Theorem 2, we see that the sample cross-covariance matrix $\mathbf{Z}$ plays an analogous role as the sample covariance matrix $\mathbf{W}$ in the training dynamics of full-batch gradient descent or mini-batch gradient descent when sampling with replacement.

---

3. For full-batch gradient descent, the corresponding requirement is $\mathrm{Range}(\mathbf{X}^\mathsf{T}) \subseteq \mathrm{Range}(\mathbf{X}^\mathsf{T}\mathbf{X})$, which always holds.

**Comparing the limiting vectors.** Furthermore, recall that the iterates of full-batch gradient descent with step size $\alpha < 2/\|n^{-1}\mathbf{X}^\mathsf{T}\mathbf{X}\|$ tend to the shifted min-norm solution

$$\widehat{\boldsymbol{\beta}}_\infty := \mathbf{P}_{\mathbf{X},0}\boldsymbol{\beta}_0 + (\mathbf{X}^\mathsf{T}\mathbf{X})^\dagger\mathbf{X}^\mathsf{T}\mathbf{y}, \tag{3.9}$$

where $\mathbf{P}_{\mathbf{X},0} := \mathbf{I} - \mathbf{X}^\dagger\mathbf{X}$ is the orthogonal projector onto $\mathrm{Null}(\mathbf{X})$, and $\|\cdot\|$ denotes the spectral norm of a matrix. From Remark 3, this is the same limit for mini-batching with replacement. In particular, note that this limit is always independent of the step size $\alpha$.

On the other hand, as a corollary of Theorem 2, we see that mini-batch gradient descent with random reshuffling, using a step size small enough so that $\|(\mathbf{I} - B\alpha\mathbf{Z})\mathbf{P}_\mathbf{Z}\| < 1$ (i.e. based on the non-zero eigenvalues of $\mathbf{Z}$), converges to a solution $\bar{\boldsymbol{\beta}}_\infty$ that can exhibit *more complex interactions between the mini-batches* and *a dependence on the step size*.

**Corollary 4 (Limit with random reshuffling)** *Consider the same setup as Theorem 2. If $\mathbf{Z}$ is positive semidefinite and $B\alpha < 2/\|n^{-1}\widetilde{\mathbf{X}}^\mathsf{T}\mathbf{X}\|$, then $\bar{\boldsymbol{\beta}}_k \to \bar{\boldsymbol{\beta}}_\infty$ as $k \to \infty$, where*

$$\bar{\boldsymbol{\beta}}_\infty \equiv \bar{\boldsymbol{\beta}}_\infty(\alpha) := \mathbf{P}_{\mathbf{Z},0}\boldsymbol{\beta}_0 + (\widetilde{\mathbf{X}}^\mathsf{T}\mathbf{X})^\dagger\widetilde{\mathbf{X}}^\mathsf{T}\mathbf{y}.$$

We can examine the two limits $\bar{\boldsymbol{\beta}}_\infty$ and $\widehat{\boldsymbol{\beta}}_\infty$ from Corollary 4 and (3.9) in the over and underparameterized regimes more carefully. For simplicity, we will assume that $\mathbf{X}$ is full rank here to avoid the complexities in the rank deficient case. Recall that $\mathbf{y} = \mathbf{X}\boldsymbol{\beta}_* + \boldsymbol{\eta}$, and since $\mathbb{R}^n = \mathrm{Range}(\mathbf{X}) \oplus \mathrm{Null}(\mathbf{X}^\mathsf{T})$, we can write $\boldsymbol{\eta} = \mathbf{X}\boldsymbol{\theta} + \boldsymbol{\xi}$ for some $\boldsymbol{\theta} \in \mathbb{R}^p$ and $\boldsymbol{\xi} \in \mathrm{Null}(\mathbf{X}^\mathsf{T})$.

- In the overparameterized regime ($p \geq n$), we have $\widehat{\boldsymbol{\beta}}_\infty = \mathbf{P}_{\mathbf{X},0}\boldsymbol{\beta}_0 + \mathbf{P}_{\mathbf{X}^\mathsf{T}}\boldsymbol{\beta}_* + \mathbf{P}_{\mathbf{X}^\mathsf{T}}\boldsymbol{\theta}$ and $\bar{\boldsymbol{\beta}}_\infty = \mathbf{P}_{\mathbf{Z},0}\boldsymbol{\beta}_0 + \mathbf{P}_\mathbf{Z}\boldsymbol{\beta}_* + \mathbf{P}_\mathbf{Z}\boldsymbol{\theta}$, since $\boldsymbol{\xi} = \mathbf{0}$ (here, $\mathbf{P}_{\mathbf{X}^\mathsf{T}}$ is the orthogonal projector onto $\mathrm{Range}(\mathbf{X}^\mathsf{T})$). Thus, if the ranges of $\mathbf{X}^\mathsf{T}$ and $\mathbf{Z}$ are close, then the two limits are also similar, regardless of the noise vector $\boldsymbol{\eta}$. Specifically, the two subspaces can be shown to coincide if $\mathrm{Range}(\mathbf{X}^\mathsf{T}) \subseteq \mathrm{Range}(\mathbf{X}^\mathsf{T}\widetilde{\mathbf{X}})$. Therefore, if $\widetilde{\mathbf{X}}$ is also full rank (which is typical), then the two limits are actually the same (in particular, the dependence of $\bar{\boldsymbol{\beta}}_\infty$ on the step size vanishes). However, we emphasize that in this case, the two *trajectories* still differ in a step size-dependent way.

- In the underparameterized regime ($p < n$), we have $\widehat{\boldsymbol{\beta}}_\infty = \boldsymbol{\beta}_* + \boldsymbol{\theta}$, but, assuming that $\widetilde{\mathbf{X}}$ is also full rank (so $\mathbf{Z} = \widetilde{\mathbf{X}}^\mathsf{T}\mathbf{X}$ is invertible), $\bar{\boldsymbol{\beta}}_\infty = \boldsymbol{\beta}_* + \boldsymbol{\theta} + (\widetilde{\mathbf{X}}^\mathsf{T}\mathbf{X})^{-1}\widetilde{\mathbf{X}}^\mathsf{T}\boldsymbol{\xi}$. Since the nullspaces of $\mathbf{X}^\mathsf{T}$ and $\widetilde{\mathbf{X}}^\mathsf{T}$ are not necessarily close (so $\widetilde{\mathbf{X}}^\mathsf{T}\boldsymbol{\xi} \neq \mathbf{0}$), we find that the two limits easily exhibit a step size-dependent difference in this case with non-zero noise $\boldsymbol{\eta}$.

**Comparing the trajectories.** Note that from Theorem 2, the error of mini-batch gradient descent with random reshuffling depends on $B\alpha\mathbf{Z}$. This can be compared with a dependence on $\alpha\mathbf{W}$ in the full-batch case. Since $\mathbf{Z}$ matches $\mathbf{W}$ up to leading order (3.3) in $\alpha$, this suggests that if a step size of $\alpha/B$ is used for mini-batch gradient descent with $B$ mini-batches, then its dynamics should be very similar to those of full-batch gradient descent with step size $\alpha$. The following remark establishes this intuition rigorously for infinitesimal step sizes.

**Remark 5 (Linear scaling and gradient flow)** *From Theorem 2, initialized at $\bar{\boldsymbol{\beta}}_{k-1}$, and using the fact that $\mathbf{Z}^\dagger\mathbf{Z}\widetilde{\mathbf{X}}^\mathsf{T} = \widetilde{\mathbf{X}}^\mathsf{T}$, the error vector of mini-batch gradient descent with random reshuffling with $B$ mini-batches and step size $\alpha/B$ satisfies*

$$\bar{\boldsymbol{\beta}}_k - \boldsymbol{\beta}_* = \left(\mathbf{I} - \frac{\alpha}{n}\widetilde{\mathbf{X}}^\mathsf{T}\mathbf{X}\right)(\bar{\boldsymbol{\beta}}_{k-1} - \boldsymbol{\beta}_*) + \frac{\alpha}{n}\widetilde{\mathbf{X}}^\mathsf{T}\boldsymbol{\eta}.$$

*By rearranging this expression, recalling that $\mathbf{Z} = \mathbf{W} + O(\alpha)$, we obtain*

$$\frac{\bar{\boldsymbol{\beta}}_k - \bar{\boldsymbol{\beta}}_{k-1}}{\alpha} = \frac{1}{n}\mathbf{X}^\mathsf{T}(\mathbf{y} - \mathbf{X}\bar{\boldsymbol{\beta}}_{k-1}) + O(\alpha).$$

*Hence, by taking the limit as $\alpha \to 0$, we deduce that the continuous dynamics correspond to the ordinary differential equation*

$$\frac{\mathrm{d}}{\mathrm{d}t}\bar{\boldsymbol{\beta}}(t) = \frac{1}{n}\mathbf{X}^T(\mathbf{y} - \mathbf{X}\bar{\boldsymbol{\beta}}(t)). \tag{3.10}$$

*This is the same differential equation for the gradient flow corresponding to full-batch gradient descent (e.g. Advani et al., 2020; Ali et al., 2019). Naturally, this also coincides with the continuous-time dynamics of the model of SGD when sampling with replacement discussed in Remark 3. As a consequence, we deduce that a gradient flow analysis cannot distinguish the effects of batching when sampling without replacement.*

**Remark 6 (Large step sizes)** *While Remark 5 shows that the dynamics of mini-batch gradient descent with random reshuffling are similar to those of full-batch gradient descent small step sizes using linear scaling, the two dynamics can be dramatically different for large step sizes. For example, if the step size satisfies $\alpha > 2/\|n^{-1}\mathbf{X}^\mathsf{T}\mathbf{X}\|$ but $B\alpha < 2/\|n^{-1}\widetilde{\mathbf{X}}^\mathsf{T}\mathbf{X}\|$, then full-batch gradient descent diverges while mini-batch gradient descent still converges. For a simple numerical demonstration of this phenomenon, see Appendix D.*

Next, a natural question is whether an explicit condition, based only on the data $\mathbf{X}$, can be formulated for how small the step size $\alpha$ needs to be for mini-batch gradient descent with random reshuffling to converge as guaranteed by Corollary 4. We can show the following sufficient condition for two-batch gradient descent: recall from Example 1 that in this setting, $\mathbf{Z} = \frac{1}{2}(\mathbf{W}_1 + \mathbf{W}_2) - \frac{1}{4}\alpha(\mathbf{W}_2\mathbf{W}_1 + \mathbf{W}_1\mathbf{W}_2)$ is a non-commutative polynomial of the mini-batch covariances $\mathbf{W}_1, \mathbf{W}_2$.

**Proposition 7** *If full-batch gradient descent with step size $2\alpha$ converges (i.e. $\alpha < 1/(n^{-1}\|\mathbf{X}^\mathsf{T}\mathbf{X}\|)$), then two-batch gradient descent with step size $\alpha$ also converges (i.e. $\|(\mathbf{I} - 2\alpha\mathbf{Z})\mathbf{P}_\mathbf{Z}\| < 1$).*

The proof of Proposition 7, which uses some matrix analysis, is given in Appendix B.2.1. To explain why this seemingly-simple statement is non-trivial, observe that it is not even immediately obvious when $\mathbf{Z}$ is positive semidefinite since it may have negative eigenvalues if $\alpha$ is large enough (unlike the covariance matrix $\mathbf{W}$). Note that the converse of Proposition 7 is not true as discussed previously in Remark 6. Furthermore, based on the correspondence with full-batch gradient descent using the linear scaling rule, Proposition 7 suggests that mini-batch gradient descent with random reshuffling has some sort of shrinkage effect on the operator norm of $\mathbf{Z}$ compared to $\mathbf{W}$.

### 3.2. Generalization error dynamics

Next, we provide an exact formula for the generalization error of the mean iterate of mini-batch gradient descent with random reshuffling, which corresponds to the usual bias-variance decomposition, The following result shows that the bias component of the generalization error (i.e. the first two terms) only depends on the sample cross-covariance matrix $\mathbf{Z}$, and the variance component (i.e. the last term) depends on $\mathbf{Z}$ and the modified features through $\widetilde{\mathbf{X}}^\mathsf{T}\widetilde{\mathbf{X}}$.

**Theorem 8** *Consider the same setup as Theorem 2. Then for all $k \geq 0$, the generalization error (2.4) of the mean mini-batch gradient descent iterate $\bar{\boldsymbol{\beta}}_k$ is given by*

$$
\begin{aligned}
R_{\mathbf{X}}(\bar{\boldsymbol{\beta}}_k) = {} & (\boldsymbol{\beta}_0 - \boldsymbol{\beta}_*)^\mathsf{T} \mathbf{P}_{\mathbf{Z},0} \Sigma \mathbf{P}_{\mathbf{Z},0} (\boldsymbol{\beta}_0 - \boldsymbol{\beta}_*) \\
& + (\boldsymbol{\beta}_0 - \boldsymbol{\beta}_*)^\mathsf{T} \mathbf{P}_{\mathbf{Z}} (\mathbf{I} - B\alpha\mathbf{Z})^k \Sigma (\mathbf{I} - B\alpha\mathbf{Z})^k \mathbf{P}_{\mathbf{Z}} (\boldsymbol{\beta}_0 - \boldsymbol{\beta}_*) \\
& + \frac{\sigma^2}{n} \operatorname{Tr}\left( \left[ \mathbf{I} - (\mathbf{I} - B\alpha\mathbf{Z})^k \right] \Sigma \left[ \mathbf{I} - (\mathbf{I} - B\alpha\mathbf{Z})^k \right] \mathbf{Z}^\dagger \left( \frac{1}{n} \widetilde{\mathbf{X}}^\mathsf{T} \widetilde{\mathbf{X}} \right) \mathbf{Z}^\dagger \right).
\end{aligned}
$$

The proof of Theorem 8, which uses the error dynamics from Theorem 2, appears in Appendix B.1.5. Theorem 8 shows that the generalization errors of mini-batch and full-batch gradient descent also correspond under the linear scaling rule, which is consistent with Remark 5 (for numerical experiments demonstrating this, see Appendix D).

As a straightforward corollary, we can also write down the limiting risk of mini-batch gradient descent with a small enough step size, complementing Corollary 4, which shows that the limiting risk consists of the constant term corresponding to $\mathbf{P}_{\mathbf{Z},0}(\boldsymbol{\beta}_0 - \boldsymbol{\beta}_*)$, the components of the initial error in the frozen subspace, and a term corresponding to overfitting the noise that is magnified by small eigenvalues of $\mathbf{Z}$.

**Corollary 9** *Consider the same setup as Theorem 8. If $\|(\mathbf{I} - B\alpha\mathbf{Z})\mathbf{P}_{\mathbf{Z}}\| < 1$, then $\bar{\boldsymbol{\beta}}_k \to \bar{\boldsymbol{\beta}}_\infty = \mathbf{P}_{\mathbf{Z},0}\boldsymbol{\beta}_0 + (\widetilde{\mathbf{X}}^\mathsf{T}\widetilde{\mathbf{X}})^\dagger \widetilde{\mathbf{X}}^\mathsf{T} \mathbf{y}$ as $k \to \infty$, and the limiting generalization error is given by*

$$
R_{\mathbf{X}}(\bar{\boldsymbol{\beta}}_\infty) = (\boldsymbol{\beta}_0 - \boldsymbol{\beta}_*)^\mathsf{T} \mathbf{P}_{\mathbf{Z},0} \Sigma \mathbf{P}_{\mathbf{Z},0} (\boldsymbol{\beta}_0 - \boldsymbol{\beta}_*) + \frac{\sigma^2}{n} \operatorname{Tr}\left( \Sigma \mathbf{Z}^\dagger \left( \frac{1}{n} \widetilde{\mathbf{X}}^\mathsf{T} \widetilde{\mathbf{X}} \right) \mathbf{Z}^\dagger \right).
$$

Note that the generalization error depends on both $\mathbf{Z}$ and the covariance of the modified features $\widetilde{\mathbf{X}}^\mathsf{T}\widetilde{\mathbf{X}}$, rather than only on $\mathbf{Z}$. If we specialize to the case of two-batch gradient descent again, then we are able to show the following result, which bounds the generalization error within a narrow interval that only depends on $\mathbf{Z}$, under a natural assumption on the step size $\alpha$ that was shown in Proposition 7 to imply convergence.

**Proposition 10** *Consider the same setup as Theorem 8 with $B = 2$. If $\alpha \leq 1/(n^{-1}\|\mathbf{X}^\mathsf{T}\mathbf{X}\|)$, then for all $k \geq 0$, $R_{\mathbf{X}}(\bar{\boldsymbol{\beta}}_k) \in [R_-(k), R_+(k)]$, where*

$$
\begin{aligned}
R_\pm(k) := {} & (\boldsymbol{\beta}_0 - \boldsymbol{\beta}_*)^\mathsf{T} \mathbf{P}_{\mathbf{Z},0} \Sigma \mathbf{P}_{\mathbf{Z},0} (\boldsymbol{\beta}_0 - \boldsymbol{\beta}_*) \\
& + (\boldsymbol{\beta}_0 - \boldsymbol{\beta}_*)^\mathsf{T} \mathbf{P}_{\mathbf{Z}} (\mathbf{I} - 2\alpha\mathbf{Z})^k \Sigma (\mathbf{I} - 2\alpha\mathbf{Z})^k \mathbf{P}_{\mathbf{Z}} (\boldsymbol{\beta}_0 - \boldsymbol{\beta}_*) \\
& + \left( 1 \pm \alpha n^{-1}\|\mathbf{X}^\mathsf{T}\mathbf{X}\| \right) \frac{\sigma^2}{n} \operatorname{Tr}\left( \left[ \mathbf{I} - (\mathbf{I} - 2\alpha\mathbf{Z})^k \right] \Sigma \left[ \mathbf{I} - (\mathbf{I} - 2\alpha\mathbf{Z})^k \right] \mathbf{Z}^\dagger \right).
\end{aligned}
$$

*The upper bound is tight if $\mathbf{W}_1 = \mathbf{W}_2 = c^2\mathbf{I}$ for some $c > 0$ and $\alpha = 2/c$. Furthermore, $\bar{\boldsymbol{\beta}}_k \to \bar{\boldsymbol{\beta}}_\infty$ as $k \to \infty$, and the limiting generalization error lies in the interval*

$$
R_{\mathbf{X}}(\bar{\boldsymbol{\beta}}_\infty) \in (\boldsymbol{\beta}_0 - \boldsymbol{\beta}_*)^\mathsf{T} \mathbf{P}_{\mathbf{Z},0} \Sigma \mathbf{P}_{\mathbf{Z},0} (\boldsymbol{\beta}_0 - \boldsymbol{\beta}_*) + (1 \pm \alpha n^{-1}\|\mathbf{X}^\mathsf{T}\mathbf{X}\|) \frac{\sigma^2}{n} \operatorname{Tr}\left( \Sigma \mathbf{Z}^\dagger \right).
$$

The proof of Proposition 10, which relies on some matrix analysis, is given in Appendix B.2.2. Note that the width of the interval is linear in the step size $\alpha$, and thus shrinks to zero as $\alpha \to 0$.

**Remark 11** *Instead of studying the generalization error of the mean iterate $\bar{\boldsymbol{\beta}}_k = \mathbb{E}_\tau[\boldsymbol{\beta}_k^{(B)}]$, it would also be of interest to understand the expected generalization error of the random iterate $\boldsymbol{\beta}_k^{(B)}$ itself; that is, $\mathbb{E}_{\tau,\mathbf{x},\boldsymbol{\eta}}[(\mathbf{x}^{\mathsf{T}}\boldsymbol{\beta}_k^{(B)} - \mathbf{x}^{\mathsf{T}}\boldsymbol{\beta}_*)^2 \mid \mathbf{X}] = \mathbb{E}_{\tau,\boldsymbol{\eta}}[\|\boldsymbol{\beta}_k^{(B)} - \boldsymbol{\beta}_*\|_\Sigma^2 \mid \mathbf{X}]$, where the expectation over the random reshuffling process is taken at the end. By a bias–variance decomposition for the norm of a random vector (e.g. [Gower and Richtárik, 2015](), Lemma 4.1), it can be shown that $\mathbb{E}_{\tau,\boldsymbol{\eta}}[\|\boldsymbol{\beta}_k^{(B)} - \boldsymbol{\beta}_*\|_\Sigma^2 \mid \mathbf{X}] = R_\mathbf{X}(\bar{\boldsymbol{\beta}}_k) + \mathbb{E}_{\tau,\boldsymbol{\eta}}[\|\boldsymbol{\beta}_k^{(B)} - \bar{\boldsymbol{\beta}}_k\|_\Sigma^2 \mid \mathbf{X}]$. Thus, our exact description of $R_\mathbf{X}(\bar{\boldsymbol{\beta}}_k)$ provides a lower bound for this notion of expected generalization error. The difference between these two quantities, $\mathbb{E}_{\tau,\boldsymbol{\eta}}[\|\boldsymbol{\beta}_k^{(B)} - \bar{\boldsymbol{\beta}}_k\|_\Sigma^2 \mid \mathbf{X}]$, corresponds to the variance of $\boldsymbol{\beta}_k^{(B)}$ over the random reshuffling process (averaged over the noise).*

### 3.3. Asymptotic analysis

In this section, we aim to provide more insights into the effects of batching without replacement by interpreting our main results on the training error (Theorem 2) and generalization error (Theorem 8) asymptotically. This will allow us to obtain a finer characterization of the sample cross-covariance matrix $\mathbf{Z}(\alpha/B) = \frac{1}{n}\widetilde{\mathbf{X}}^{\mathsf{T}}\mathbf{X}$, which is quite non-trivial to analyze in general as it is a non-commutative polynomial of the mini-batch covariance matrices, and compare it with the sample covariance matrix $\mathbf{W} = \frac{1}{n}\mathbf{X}^{\mathsf{T}}\mathbf{X}$, its full-batch analogue (under linear scaling).

#### 3.3.1. ASYMPTOTIC ANALYSIS IN THE LARGE $n$, FIXED $p$ REGIME

We begin by considering the more classical statistical regime where $p$ is fixed and $n \to \infty$. Since we assume that the features $\mathbf{x}_i$ are i.i.d. with $\mathbb{E}\left[\mathbf{x}_i\mathbf{x}_i^{\mathsf{T}}\right] = \Sigma$, by the law of large numbers, the sample covariances $\mathbf{W} = \frac{1}{n}\mathbf{X}^{\mathsf{T}}\mathbf{X}$ and $\mathbf{W}_b = \frac{B}{n}\mathbf{X}_b^{\mathsf{T}}\mathbf{X}_b$, $b = 1, \ldots B$ tend to $\Sigma$ as $n \to \infty$, almost surely. Therefore, by independence, $\mathbf{Z}(\alpha/B)$ tends to $\Sigma(\mathbf{I} - p_{B,\alpha}(\Sigma))$, where $p_{B,\alpha}$ is a certain polynomial that depends on the number of mini-batches $B$ and step size $\alpha$. If we denote the eigenvalues of $\Sigma$ by $\lambda_i$, then the limiting eigenvalues of $\mathbf{Z}(\alpha/B)$ are given by $\lambda_i(1 - p_{B,\alpha}(\lambda_i))$.

For example, if $B = 2$, then $\mathbf{Z}(\alpha/2) = \frac{1}{2}(\mathbf{W}_1 + \mathbf{W}_2) - \frac{1}{8}\alpha(\mathbf{W}_2\mathbf{W}_1 + \mathbf{W}_1\mathbf{W}_2)$ converges to $\Sigma - \frac{1}{4}\alpha\Sigma^2$, so $p_{2,\alpha}(\Sigma) = \frac{1}{4}\alpha\Sigma$. In particular, note that the limiting spectrum of $\mathbf{W}$ is shrunk compared to $\mathbf{Z}$.[4] In general, for any $B$, we have the following expression for $p_{B,\alpha}$:

**Proposition 12** *Suppose that $p$ is fixed. Then as $n \to \infty$, $\mathbf{Z}(\alpha/B) \to \Sigma(\mathbf{I} - p_{B,\alpha}(\Sigma))$ almost surely, where*

$$p_{B,\alpha}(\Sigma) = \sum_{i=1}^{B-1}(-1)^{i+1}\frac{(B-1)!(B-1-i)!}{(i+1)!}\left(\frac{\alpha}{B}\right)^i \Sigma^i.$$

The proof of this result uses the algebraic representation of $\Pi_b$ as a function of all the other mini-batches to write down the limit of each $\mathbf{W}_b\Pi_b$, which allows for the limit of $\mathbf{Z} = \frac{1}{B}\sum_{b=1}^{B}\mathbf{W}_b\Pi_b$ to be obtained by symmetry. For the details, see Appendix B.3.1.

Observe that Proposition 12 implies that the sample cross-covariance $\mathbf{Z}$ is not a consistent estimator of the true (uncentered) covariance matrix $\Sigma$ of the features, unlike $\mathbf{W}$. Moreover, we see that although $\mathbf{Z}$ matches $\mathbf{W}$ up to leading order in $\alpha$, asymptotically, batching results in a step size-dependent shrinkage of the spectrum of $\mathbf{W}$ (for small enough $\alpha$ satisfying, say, $\alpha\|\mathbf{W}\| \leq 1$).

---

4. For another example, if $B = 3$, then $\mathbf{Z}(\alpha/3) = \frac{1}{3}(\mathbf{W}_1 + \mathbf{W}_2 + \mathbf{W}_3) - \frac{1}{18}\alpha(\mathbf{W}_1\mathbf{W}_2 + \mathbf{W}_1\mathbf{W}_3 + \mathbf{W}_2\mathbf{W}_1 + \mathbf{W}_2\mathbf{W}_3 + \mathbf{W}_3\mathbf{W}_1 + \mathbf{W}_3\mathbf{W}_2) + \frac{1}{162}\alpha^2(\mathbf{W}_1\mathbf{W}_2\mathbf{W}_3 + \mathbf{W}_1\mathbf{W}_3\mathbf{W}_2 + \mathbf{W}_2\mathbf{W}_1\mathbf{W}_3 + \mathbf{W}_2\mathbf{W}_3\mathbf{W}_1 + \mathbf{W}_3\mathbf{W}_1\mathbf{W}_2 + \mathbf{W}_3\mathbf{W}_2\mathbf{W}_1)$, which converges to $\Sigma - \frac{1}{3}\alpha\Sigma^2 + \frac{1}{27}\alpha^2\Sigma^3$, so $p_{3,\alpha}(\Sigma) = \frac{1}{3}\alpha\Sigma - \frac{1}{27}\alpha^2\Sigma^2$.

The key idea behind Proposition 12 is that by exploiting the independence of each mini-batch and the algebraic properties of $\Pi_b$, the matrix $\Pi_b$ that modifies each mini-batch turns out to be asymptotically independent of $b$ as $n \to \infty$; in fact, the limit of each $\Pi_b$ is exactly the matrix $\mathbf{I} - p_{B,\alpha}(\Sigma)$ from Proposition 12. We can take this idea and make it an explicit assumption (justified by the fact that it holds asymptotically) to elaborate on the implications of batching by providing an *explicit description* of how the *trajectories* of mini-batch gradient descent with random reshuffling differ from the full-batch case under linear scaling.

**Proposition 13** *Suppose that $\Pi_b = \Pi := \mathbf{I} - p(\mathbf{W})$ for each $b = 1, \ldots, B$, where $p \equiv p_\alpha$ is some polynomial, and that $\mathbf{X}$ and $\Pi$ are invertible. Let $\mathbf{X} = \mathbf{USV}^\mathsf{T}$ be a singular value decomposition of $\mathbf{X}$, so that $\mathbf{W} = \mathbf{V}(\frac{1}{n}\mathbf{S}^\mathsf{T}\mathbf{S})\mathbf{V}^\mathsf{T}$ where $\frac{1}{n}\mathbf{S}^\mathsf{T}\mathbf{S}$ is a diagonal matrix with (non-zero) eigenvalues denoted by $\hat{\lambda}_1, \ldots, \hat{\lambda}_p$. Then for $i = 1, \ldots, p$, the $i$th coordinate (in the eigenbasis $\mathbf{V}$) of the error vector $\bar{\boldsymbol{\beta}}_k - \boldsymbol{\beta}_*$ after $k$ epochs of mini-batch gradient descent is given by*

$$
\begin{aligned}
[\mathbf{V}^\mathsf{T}(\bar{\boldsymbol{\beta}}_k - \boldsymbol{\beta}_*)]_i = {} & [1 - \alpha\hat{\lambda}_i(1 - p(\hat{\lambda}_i))]^k[\mathbf{V}^\mathsf{T}(\boldsymbol{\beta}_0 - \boldsymbol{\beta}_*)]_i \\
& + \frac{1}{\hat{\lambda}_i}\left(1 - [1 - \alpha\hat{\lambda}_i(1 - p(\hat{\lambda}_i))]^k\right)[\mathbf{U}^\mathsf{T}\boldsymbol{\eta}]_i.
\end{aligned}
\tag{3.11}
$$

*If we make the further simplifying assumption that $\mathbf{V}^\mathsf{T}\Sigma\mathbf{V} = \Lambda$ is diagonal with eigenvalues $\lambda_1, \ldots, \lambda_p$,[5] then the corresponding generalization error is given by*

$$
\begin{aligned}
R_{\mathbf{X}}(\bar{\boldsymbol{\beta}}_k) = {} & \sum_{i=1}^{p} \lambda_i[1 - \alpha\hat{\lambda}_i(1 - p(\hat{\lambda}_i))]^{2k}[\mathbf{V}^\mathsf{T}(\boldsymbol{\beta}_0 - \boldsymbol{\beta}_*)]_i \\
& + \frac{\sigma^2}{n}\sum_{i=1}^{p}\frac{\lambda_i}{\hat{\lambda}_i}\left(1 - [1 - \alpha\hat{\lambda}_i(1 - p(\hat{\lambda}_i))]^k\right)^2.
\end{aligned}
\tag{3.12}
$$

The proof of these claims is obtained from the dynamics described in Theorems 2 and 8 under the specific assumptions imposed, and can be found in Appendix B.3.2.

For comparison, the corresponding quantities for full-batch gradient descent are the same as those in Proposition 13 with $\hat{\lambda}_i(1 - p_{B,\alpha}(\hat{\lambda}_i))$ replaced by $\hat{\lambda}_i$. Therefore, if we take $p = p_{B,\alpha}$ from Proposition 12 as the specific polynomial that motivated the setup, then we deduce that the convergence rate $[1 - \alpha\hat{\lambda}_i(1 - p(\hat{\lambda}_i))]$ for mini-batch gradient descent (3.11) is comparatively smaller due to the shrinkage effect on the spectrum of $\mathbf{W}$. The overall impact on the generalization error (3.12) is less clear, since the change in the convergence rate implies a different tradeoff between fitting the signal (i.e. bias) and the noise (i.e. variance). However, if early stopping is used to minimize the generalization error (e.g. Sonthalia et al., 2024), then a particular consequence is mini-batch gradient descent with random reshuffling may have a *different optimal stopping time and a different early stopped risk (possibly lower)*, compared with full-batch gradient descent.

### 3.3.2. ASYMPTOTIC ANALYSIS IN THE PROPORTIONAL REGIME: LARGE $n$ AND $p$

Next, we consider the proportional regime in which both $n, p \to \infty$ such that $p/n \to \gamma \in (0, \infty)$. This setting has been extensively studied in the context of modern large-scale machine learning in prior theoretical works (e.g. Hastie et al., 2022; Couillet and Liao, 2022; Mei and Montanari,

---

5. For example, this holds if we assume that the features $\mathbf{x}_i$ are isotropic so that $\Sigma$ is a scalar multiple of the identity.

2022; Ba et al., 2022; Wang et al., 2024). In this regime, the sample covariance $\mathbf{W}$ does not have a deterministic limit in general. However, its limiting spectral distribution can be studied using tools from random matrix theory if assume that the features $\mathbf{x}_i$ satisfy some concentration properties.

We will also consider the more tractable setting of two-batch gradient descent, recalling that $\alpha \mathbf{Z}(\alpha/2) = \frac{1}{2}\alpha(\mathbf{W}_1 + \mathbf{W}_2) - \frac{1}{8}\alpha^2(\mathbf{W}_2\mathbf{W}_1 + \mathbf{W}_1\mathbf{W}_2)$. This case is also already difficult to analyze in the proportional regime since it requires finding a non-trivial limiting distribution of a non-commutative polynomial of random matrices.

For the remainder of this section, we will assume that the entries of $\mathbf{x}_i$ are i.i.d. standard Gaussian.[6] In this case, it is well-known (Marčenko and Pastur, 1967; Bai and Silverstein, 2010) that almost surely, the empirical spectral distribution[7] $F_{\alpha\mathbf{W}}(x)$ of $\alpha\mathbf{W}$ (known as a *Wishart matrix*) converges in distribution to the *Marchenko-Pastur distribution* with ratio parameter $\gamma$ and variance $\alpha$, which has probability measure $\nu_{\gamma,\alpha}$ given by

$$\mathrm{d}\nu_{\gamma,\alpha}(x) := \frac{1}{2\pi\alpha\gamma x}\sqrt{(x_+ - x)(x - x_-)} + \left(1 - \frac{1}{\gamma}\right)_+ \mathbb{1}_{\{x=0\}}, \quad \text{where } x_\pm := \alpha(1 \pm \sqrt{\gamma})^2.$$

That is, $\nu_{\gamma,\alpha}$ has a density supported on $[x_-, x_+]$, and a point mass of $(1 - \gamma^{-1})$ at zero if and only if $\gamma > 1$ (i.e. in the overparameterized regime).

To understand the limiting spectrum of $\alpha\mathbf{Z}(\alpha/2)$, our starting point is the observation that

$$\alpha\mathbf{Z}(\alpha/2) = \frac{\alpha}{2}(\mathbf{W}_1 + \mathbf{W}_2) - \frac{\alpha^2}{8}(\mathbf{W}_2\mathbf{W}_1 + \mathbf{W}_2\mathbf{W}_1) = p\left(\frac{\alpha}{2}\mathbf{W}_1, \frac{\alpha}{2}\mathbf{W}_2\right) \tag{3.13}$$

is a *non-commutative polynomial* $p(x,y) = x + y - \frac{1}{2}(xy + yx)$ *in the independent Wishart matrices* $\frac{\alpha}{2}\mathbf{W}_1$ *and* $\frac{\alpha}{2}\mathbf{W}_2$. To understand its spectrum, we need tools from free probability theory, which, roughly speaking, deals with a notion of *free independence* for non-commutative random variables: for the precise mathematical setup, we refer to a standard reference, e.g. Mingo and Speicher (2017).

The key result needed is that under the Gaussian assumption on $\mathbf{x}_i$, $\frac{\alpha}{2}\mathbf{W}_1$ and $\frac{\alpha}{2}\mathbf{W}_2$ are asymptotically free (Mingo and Speicher, 2017, Section 4.5.1), which implies that the limiting spectral distribution of $\alpha\mathbf{Z}(\alpha/2)$ is the spectral distribution of the polynomial $p(w_1, w_2)$ of two freely independent Marchenko-Pastur distributions $w_1, w_2$ with ratio parameter $2\gamma$ and variance $\alpha/2$.

*However, there is no closed-form or convenient analytical expression for characterizing the limiting spectral distribution of* $\alpha\mathbf{Z}(\alpha/2)$. Instead, we were able to use a general algorithm for computing the spectral distribution of a polynomial of free random variables from Belinschi et al. (2017) for this task by lifting to the space of *operator-valued random variables*. Specifically, after developing a linearization of the non-commutative polynomial in (3.13), we implemented the algorithm in Belinschi et al. (2017) to compute the operator-valued Stieltjes transform of the linearization, from which we could numerically extract the desired spectral distribution of $p(w_1, w_2)$. For a detailed description of our procedure, see Appendix C.

Figure 1 demonstrates our results from computing the limiting spectral distributions of $\alpha\mathbf{Z}(\alpha/2)$ and $\alpha\mathbf{W}$ in the underparameterized ($\gamma < 1$) and overparameterized ($\gamma > 1$) regimes. We observe

---

6. While the limiting spectrum of $\mathbf{W}$ can be described under more general models, such as assuming that $\mathbf{x}_i = \Sigma^{1/2}\mathbf{z}_i$ for some $\mathbf{z}_i$ with i.i.d. coordinates (Dobriban and Wager, 2018; Hastie et al., 2022), or that $\mathbf{x}_i$ is a random vector that is subgaussian or satisfies convex concentration (Couillet and Liao, 2022), we will require this strong assumption to study the limiting spectrum of $\mathbf{Z}$ using tools from free probability theory.

7. The empirical spectral distribution of a symmetric matrix $\mathbf{A} \in \mathbb{R}^{p \times p}$ with eigenvalues $\lambda_i(\mathbf{A})$ is defined by $F_\mathbf{A}(x) := \frac{1}{p}\sum_{i=1}^p \mathbb{1}_{\{\lambda_i(\mathbf{A}) \leq x\}}$.

that batching results in a non-linear shrinkage effect of the spectrum of $\alpha\mathbf{W}$. This is consistent with the conclusions from Section 3.3.1 in a different asymptotic regime (which does not allow for the overparameterized case). The close adherence between the theoretical predictions and simulations with moderately-sized matrices also highlights the predictive capacity of the asymptotic theory.

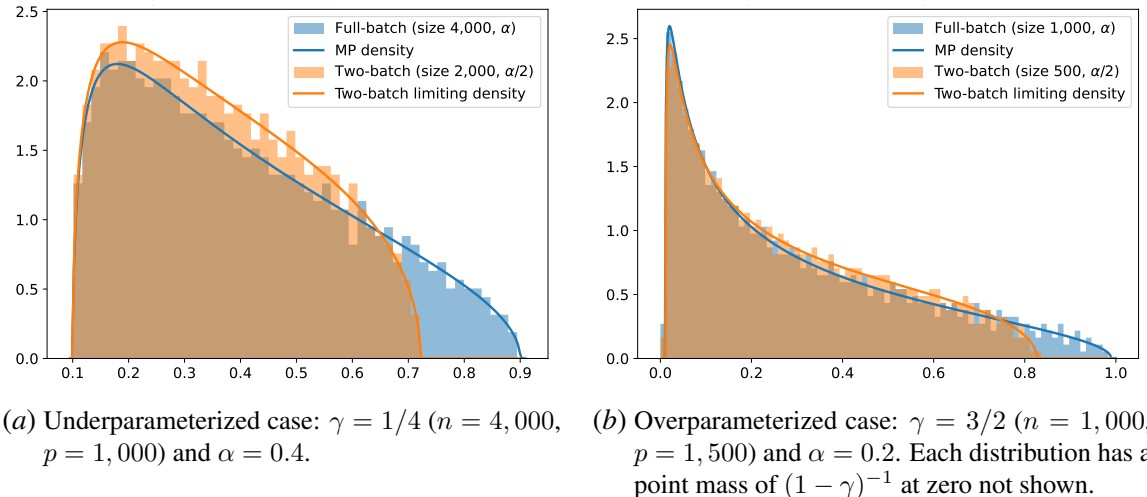

(a) Underparameterized case: $\gamma = 1/4$ ($n = 4,000$, $p = 1,000$) and $\alpha = 0.4$.

(b) Overparameterized case: $\gamma = 3/2$ ($n = 1,000$, $p = 1,500$) and $\alpha = 0.2$. Each distribution has a point mass of $(1-\gamma)^{-1}$ at zero not shown.

Figure 1: Limiting spectral distributions (lines) of $\alpha\mathbf{W}$ (full-batch) and $\alpha\mathbf{Z}(\alpha/2)$ (two-batch) compared with empirical distribution of a single $n \times p$ standard Gaussian matrix (histogram).

## 4. Conclusion

In this work, we showed that the training and generalization error dynamics of mini-batch gradient descent with random reshuffling for least squares regression depend on a sample cross-covariance matrix $\mathbf{Z}$ between the original features and a set of new features that have been modified by the other mini-batches. Using this connection, we established that while the linear scaling rule for the step size matches the dynamics of mini-batch and full-batch gradient descent up to leading order, sampling without replacement results in subtle differences that a continuous-time gradient flow analysis cannot detect. We demonstrated that asymptotically, batching leads to non-linear shrinkage effects on the spectrum of the sample covariance matrix $\mathbf{W}$, which directly affects the mini-batch error dynamics.

Some future directions include studying the dynamics of mini-batch gradient descent with random reshuffling under more specific assumptions to gain insights into the optimal choice of batch size and learning rate for generalization, as well as generalizing the results to more realistic models such as one-layer networks with non-linearities. Finally, there are some random matrix questions on better understanding $\mathbf{Z}$ asymptotically in the proportional regime, such as obtaining precise analytical expressions in the Gaussian setting.

## Acknowledgments

We would like to thank the anonymous reviewers for providing additional references that have improved our discussion of the relevant literature, as well as an anonymous reviewer for pointing out a correction in a proof. ER and JL were supported by funds from NSF-DMS #2309685.

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

## Appendices

The organization of the appendices is as follows:

- Appendix A provides additional background on the error dynamics of full-batch gradient descent that are known from the literature.

- Appendix B contains the technical proofs for our results on mini-batch gradient descent with random reshuffling.

- Appendix C provides a high-level overview and details on our implementation of the algorithm from Belinschi et al. (2017) for calculating the spectral distribution of a self-adjoint polynomial of free random variables.

- Appendix D presents some additional numerical experiments.

## Appendix A. Full-batch gradient descent

In this section, we state formulas for the error dynamics and generalization error of full-batch gradient descent (i.e. with $B = 1$). These results are not novel, having appeared in the literature in many varying forms (e.g. Ali et al., 2019; Richards et al., 2022). However, they are helpful for the purposes of comparison with analogous results for mini-batch gradient descent with random reshuffling.

The first lemma gives an exact expression for the error vector that is driven by the sample covariance matrix $\mathbf{W} := \frac{1}{n}\mathbf{X}^\mathsf{T}\mathbf{X}$ of the features (i.e. Hessian of the least squares problem).

**Lemma 14** *Let $(\boldsymbol{\beta}_k)_{k\geq 0}$ be the sequence of full-batch gradient descent iterates for the least squares problem with step size $\alpha \geq 0$ and initialization $\boldsymbol{\beta}_0 \in \mathbb{R}^p$. Then for all $k \geq 0$,*

$$\boldsymbol{\beta}_k - \boldsymbol{\beta}_* = (\mathbf{I} - \alpha\mathbf{W})^k (\boldsymbol{\beta}_0 - \boldsymbol{\beta}_*) + \frac{1}{n}\left[\mathbf{I} - (\mathbf{I} - \alpha\mathbf{W})^k\right]\mathbf{W}^\dagger\mathbf{X}^\mathsf{T}\boldsymbol{\eta}. \tag{A.1}$$

*Furthermore, if $\mathbf{P}_{\mathbf{X},0} := \mathbf{I} - (\mathbf{X}^\mathsf{T}\mathbf{X})^\dagger(\mathbf{X}^\mathsf{T}\mathbf{X})$ and $\mathbf{P}_{\mathbf{X}^\mathsf{T}} := \mathbf{I} - \mathbf{P}_0$ denote the orthogonal projectors onto the nullspace and row space of $\mathbf{X}$ respectively, then we may decompose the first term as*

$$(\mathbf{I} - \alpha\mathbf{W})^k (\boldsymbol{\beta}_0 - \boldsymbol{\beta}_*) = \mathbf{P}_{\mathbf{X},0}(\boldsymbol{\beta}_0 - \boldsymbol{\beta}_*) + (\mathbf{I} - \alpha\mathbf{W})^k \mathbf{P}_{\mathbf{X}^\mathsf{T}}(\boldsymbol{\beta}_0 - \boldsymbol{\beta}_*). \tag{A.2}$$

**Proof** Since $\mathbf{y} = \mathbf{X}\boldsymbol{\beta}_* + \boldsymbol{\eta}$, the error vector satisfies the recursive relationship

$$\boldsymbol{\beta}_k - \boldsymbol{\beta}_* = \left(\mathbf{I} - \frac{\alpha}{n}\mathbf{X}^\mathsf{T}\mathbf{X}\right)(\boldsymbol{\beta}_{k-1} - \boldsymbol{\beta}_*) + \frac{\alpha}{n}\mathbf{X}^\mathsf{T}\boldsymbol{\eta}.$$

By recursively applying this relationship, and instating the definition of $\mathbf{W} = \frac{1}{n}\mathbf{X}^\mathsf{T}\mathbf{X}$, we obtain

$$\boldsymbol{\beta}_k - \boldsymbol{\beta}_* = (\mathbf{I} - \alpha\mathbf{W})^k (\boldsymbol{\beta}_0 - \boldsymbol{\beta}_*) + \frac{\alpha}{n}\sum_{j=1}^k (\mathbf{I} - \alpha\mathbf{W})^{k-j}\mathbf{X}^\mathsf{T}\boldsymbol{\eta}.$$

The proof of (A.1) is completed by using the following identity to simplify the expression for the sum above, which follows from considering the eigendecomposition of the symmetric matrix $\mathbf{X}$:

$$\sum_{j=1}^k (\mathbf{I} - \alpha\mathbf{W})^{k-j}\mathbf{X}^\mathsf{T} = \left[\mathbf{I} - (\mathbf{I} - \alpha\mathbf{W})^k\right](\alpha\mathbf{W})^\dagger\mathbf{X}^\mathsf{T}.$$

Finally, by incorporating the decomposition of the initial error

$$\boldsymbol{\beta}_0 - \boldsymbol{\beta}_* = \mathbf{P}_{\mathbf{X},0}(\boldsymbol{\beta}_0 - \boldsymbol{\beta}_*) + \mathbf{P}_{\mathbf{X}^\mathsf{T}}(\boldsymbol{\beta}_0 - \boldsymbol{\beta}_*),$$

noting that $(\mathbf{I} - \alpha\mathbf{W})^k\mathbf{P}_{\mathbf{X},0} = \mathbf{P}_{\mathbf{X},0}$, we obtain (A.2). ∎

The following lemma gives a formula for the generalization error of full-batch gradient descent, corresponding to the usual bias-variance decomposition. It reveals that the generalization error is characterized by the *eigenvalue spectrum of the sample covariance matrix* $\mathbf{W}$, the *alignment of the initial error $\boldsymbol{\beta}_0 - \boldsymbol{\beta}_*$ with the eigenspaces of* $\mathbf{W}$, as well as the *covariance of the features* $\Sigma$.

**Lemma 15** *Consider the same setup as Lemma [14]. Then for all $k \geq 0$, the generalization error ((2.4)) of the full-batch gradient descent iterates $\boldsymbol{\beta}_k$ is given by*

$$
\begin{aligned}
R_{\mathbf{X}}(\boldsymbol{\beta}_k) &= (\boldsymbol{\beta}_0 - \boldsymbol{\beta}_*)^\mathsf{T} \mathbf{P}_{\mathbf{X},0} \Sigma \mathbf{P}_{\mathbf{X},0} (\boldsymbol{\beta}_0 - \boldsymbol{\beta}_*) \\
&\quad + (\boldsymbol{\beta}_0 - \boldsymbol{\beta}_*)^\mathsf{T} \mathbf{P}_{\mathbf{X}^\mathsf{T}} (\mathbf{I} - \alpha \mathbf{W})^k \Sigma (\mathbf{I} - \alpha \mathbf{W})^k \mathbf{P}_{\mathbf{X}^\mathsf{T}} (\boldsymbol{\beta}_0 - \boldsymbol{\beta}_*) \\
&\quad + \frac{\sigma^2}{n} \operatorname{Tr} \left( \left[ \mathbf{I} - (\mathbf{I} - \alpha \mathbf{W})^k \right] \Sigma \left[ \mathbf{I} - (\mathbf{I} - \alpha \mathbf{W})^k \right] \mathbf{W}^\dagger \right).
\end{aligned}
$$

**Proof** Note that $\|\boldsymbol{\beta}_k - \boldsymbol{\beta}_*\|_\Sigma^2 = \|\Sigma^{1/2}(\boldsymbol{\beta}_k - \boldsymbol{\beta}_*)\|_2^2$, where $\|\cdot\|_2$ is the usual $\ell_2$ norm. Hence, we may expand the square in (A.1) of Lemma 14, and use the fact that the cross-terms with a linear dependence on the mean-zero noise term $\boldsymbol{\eta}$ vanish upon taking expectation. The first term of this expansion, combined with the decomposition of the initial error in (A.2), yields the first two terms of the claimed generalization error, corresponding to the bias. The remaining variance term follows from writing the second term of the expansion as a trace (i.e. writing $\|\mathbf{z}^\mathsf{T}\mathbf{z}\|_2^2 = \operatorname{Tr}(\mathbf{z}\mathbf{z}^\mathsf{T})$), using the fact that $\mathbb{E}[\boldsymbol{\eta}\boldsymbol{\eta}^\mathsf{T}] = \sigma^2 \mathbf{I}$, the cyclic property of trace, and the property $\mathbf{W}^\dagger \mathbf{W} \mathbf{W}^\dagger = \mathbf{W}^\dagger$ of the pseudoinverse. ∎

Observe that by taking the limit as $k \to \infty$ with a small enough step size, Lemma 14 shows that gradient descent converges to the min-norm solution $(\mathbf{X}^\mathsf{T}\mathbf{X})^\dagger \mathbf{X}^\mathsf{T}\mathbf{y}$ of the least squares problem, shifted by the projection of $\boldsymbol{\beta}_0$ onto the nullspace of $\mathbf{X}$. Additionally, Lemma 15 shows that the resulting generalization error is increased by small eigenvalues of $\mathbf{W}$, which corresponds to overfitting the noise.

**Corollary 16** *Consider the same setup as Lemma [14]. Let $\boldsymbol{\beta}_\infty := \mathbf{P}_{\mathbf{X},0}\boldsymbol{\beta}_0 + (\mathbf{X}^\mathsf{T}\mathbf{X})^\dagger \mathbf{X}^\mathsf{T}\mathbf{y}$. If $\alpha < 2/(n^{-1}\|\mathbf{X}^\mathsf{T}\mathbf{X}\|)$, then $\boldsymbol{\beta}_k \to \boldsymbol{\beta}_\infty$ as $k \to \infty$, and the limiting generalization error is given by*

$$
R_{\mathbf{X}}(\boldsymbol{\beta}_\infty) = (\boldsymbol{\beta}_0 - \boldsymbol{\beta}_*)^\mathsf{T} \mathbf{P}_{\mathbf{X},0} \Sigma \mathbf{P}_{\mathbf{X},0} (\boldsymbol{\beta}_0 - \boldsymbol{\beta}_*) + \frac{\sigma^2}{n} \operatorname{Tr} \left( \Sigma \mathbf{W}^\dagger \right).
$$

## Appendix B. Technical proofs for mini-batch gradient descent

In this section, we provide the technical proofs for our results on mini-batch gradient descent with random reshuffling. Specifically, in Appendix B.1, we prove the general results for general mini-batch gradient descent (Lemma 1, Theorem 2, and Theorem 8). In Appendix B.2, we prove some more precise results in the specific setting of two-batch gradient descent with $B = 2$ (Propositions 7 and 10). Finally, in Appendix B.3, we prove the results obtained in the asymptotic regime as $n \to \infty$ with fixed $p$ (Propositions 12 and 13).

### B.1. Mini-batch gradient descent

As a reminder of our setup, recall that we have partitioned the data matrix $\mathbf{X} \in \mathbb{R}^{m \times n}$ into $B$ equally-sized batches $\mathbf{X}_1, \ldots, \mathbf{X}_B \in \mathbb{R}^{(m/B) \times n}$, and we denote the corresponding sample covariance matrix of $\mathbf{X}_b$ by $\mathbf{W}_b = \frac{B}{n} \mathbf{X}_b^\mathsf{T} \mathbf{X}_b$. The modified mini-batches are defined by $\widetilde{\mathbf{X}}_b := \mathbf{X}_b \Pi_b$, where $\Pi_b$ was defined in (3.1) as

$$
\Pi_b = \frac{1}{B!} \sum_{\tau \in S_B} \prod_{j : j < \tau^{-1}(b)} (\mathbf{I} - \alpha \mathbf{W}_{\tau(j)}). \tag{B.1}
$$

Another equivalent expression for $\Pi_b$ is the following:

$$\Pi_b = \frac{1}{B!} \sum_{\tau \in S_B} \prod_{j:j>\tau^{-1}(b)} (\mathbf{I} - \alpha \mathbf{W}_{\tau(j)}). \tag{B.2}$$

This is because each summand corresponding to a permutation $\tau$ in (B.1) can be matched one-to-one to a summand corresponding to a permutation $\tau'$ in (B.2) by swapping the sub-permutations to the left and right of the batch $b$ (with position $\tau^{-1}(b)$). For example, without loss of generality, consider $\tau = (1, \ldots, b-1, b, b+1, \ldots, B)$, and let $\tau' = (B, \ldots, b+1, b, 1, \ldots, b-1)$. Then the summand in (B.1) for $\tau$ is $(\mathbf{I} - \alpha \mathbf{W}_{b-1}) \ldots (\mathbf{I} - \alpha \mathbf{W}_1)$, which exactly corresponds to the summand for $\tau'$ in (B.2).

Furthermore, by expanding, it can be verified that another expression for $\Pi_b$ is the following:[8]

$$\Pi_b = \mathbf{I} - \sum_{i=1}^{B-1} \left\{ \frac{(-1)^{i+1}}{(i+1)!} \cdot \alpha^i \sum_{\substack{\{b_1,\ldots,b_i\} \subseteq [B]\setminus\{b\} \\ b_1,\ldots,b_i \text{ distinct, ordered}}} \mathbf{W}_{b_1} \ldots \mathbf{W}_{b_i} \right\}. \tag{B.3}$$

Finally, $\widetilde{\mathbf{X}}$ denotes the concatenation of the $B$ modified mini-batches in the same order, and $\mathbf{Z} = \frac{1}{n}\widetilde{\mathbf{X}}^\mathsf{T}\mathbf{X} = \frac{1}{n}\sum_{b=1}^{B} \Pi_b \mathbf{X}_b^\mathsf{T}\mathbf{X}_b$ was defined in (3.2) to be the sample cross-covariance matrix of the modified features with the original features.

### B.1.1. PROOF OF LEMMA 1

Since $\mathbf{Z} = \frac{1}{n}\widetilde{\mathbf{X}}^\mathsf{T}\mathbf{X} = \frac{1}{n}\sum_{b=1}^{B} \widetilde{\mathbf{X}}_b^\mathsf{T}\mathbf{X}_b = \frac{1}{n}\sum_{b=1}^{B} \Pi_b \mathbf{X}_b^\mathsf{T}\mathbf{X}_b$, it suffices to show that

$$\sum_{b=1}^{B} \Pi_b \mathbf{W}_b = \sum_{b=1}^{B} \mathbf{W}_b \Pi_b$$

to prove that $\mathbf{Z}$ is symmetric, where $\Pi_b$ is defined as in (B.2). Fix $b \in \{1, \ldots, B\}$. Note that $\Pi_b \mathbf{W}_b$ and $\mathbf{W}_b \Pi_b$ are polynomials in the non-commuting variables $\mathbf{W}_1, \ldots, \mathbf{W}_B$, and that $\Pi_b$ does not contain the term $\mathbf{W}_b$. Hence, it suffices to argue that the word ending in $\mathbf{W}_b$ on the left hand side (i.e. $\Pi_b \mathbf{W}_b$) matches the word ending in $\mathbf{W}_b$ on the right hand side (i.e. the sum of the words ending in $\mathbf{W}_b$ in $\sum_{j \neq b} \mathbf{W}_j \Pi_j$).

Observe that $\Pi_b \mathbf{W}_b$ is a sum of words of the form $a_{i_1,\ldots,i_\ell} \mathbf{W}_{i_1} \mathbf{W}_{i_2} \cdots \mathbf{W}_{i_\ell} \mathbf{W}_b$, where each of the indices are distinct and $a_{i_1,\ldots,i_\ell} \in \mathbb{R}$ is a constant. From the form of $\Pi_b$, this term arises as a sum over permutations $\tau$ from a set, say $\mathcal{T} \equiv \mathcal{T}_{i_1,\ldots,i_\ell}$, such that $\tau^{-1}(b) < \tau^{-1}(i_\ell) < \cdots < \tau^{-1}(i_2) < \tau^{-1}(i_1)$:

$$a_{i_1,\ldots,i_\ell} \mathbf{W}_{i_1} \mathbf{W}_{i_2} \cdots \mathbf{W}_{i_\ell} \mathbf{W}_b = \left( \frac{1}{B!} \sum_{\tau \in \mathcal{T}} (-\alpha)^\ell \mathbf{W}_{i_1} \mathbf{W}_{i_2} \cdots \mathbf{W}_{i_\ell} \right) \mathbf{W}_b.$$

---

8. For example: with $B = 2$, $\Pi_1 = \mathbf{I} - \frac{1}{2}\alpha \mathbf{W}_2$; with $B = 3$, $\Pi_1 = \mathbf{I} - \frac{1}{2}\alpha(\mathbf{W}_2 + \mathbf{W}_3) + \frac{1}{6}\alpha^2(\mathbf{W}_2\mathbf{W}_3 + \mathbf{W}_3\mathbf{W}_2)$; with $B = 4$, $\Pi_1 = \mathbf{I} - \frac{1}{2}\alpha(\mathbf{W}_2 + \mathbf{W}_3 + \mathbf{W}_4) + \frac{1}{6}\alpha^2(\mathbf{W}_2\mathbf{W}_3 + \mathbf{W}_2\mathbf{W}_4 + \mathbf{W}_3\mathbf{W}_2 + \mathbf{W}_3\mathbf{W}_4 + \mathbf{W}_4\mathbf{W}_2 + \mathbf{W}_4\mathbf{W}_3) - \frac{1}{24}\alpha^3(\mathbf{W}_2\mathbf{W}_3\mathbf{W}_4 + \mathbf{W}_2\mathbf{W}_4\mathbf{W}_3 + \mathbf{W}_3\mathbf{W}_2\mathbf{W}_4 + \mathbf{W}_3\mathbf{W}_4\mathbf{W}_2 + \mathbf{W}_4\mathbf{W}_2\mathbf{W}_3 + \mathbf{W}_4\mathbf{W}_3\mathbf{W}_2)$; and so on.

The same word arises in the expression $\sum_{j \neq b} \mathbf{W}_j \Pi_j$ from the single term $\mathbf{W}_{i_1} \Pi_{i_1}$ with $\mathbf{W}_{i_1}$ as the leftmost matrix in the product. For each $\tau \in \mathcal{T}$, consider shifting the sub-permutation $(b, i_\ell, \ldots, i_2, i_1)$ in $\tau$ cyclically to the right (keeping the other entries fixed) to obtain the permutation $\tau'$ with sub-permutation $(i_1, b, i_\ell, \ldots, i_2)$. If $\mathcal{T}'$ denotes the set of permutations obtained from $\mathcal{T}$ in this way, then by summing over all $\tau' \in \mathcal{T}$ in $\Pi_{i_1}$ — choosing the term $-\alpha \mathbf{W}_{\tau'(j)}$ for each $j \in \{\tau^{-1}(b), \tau^{-1}(i_\ell), \ldots, \tau^{-1}(i_2)\}$, and $\mathbf{I}$ for the rest of the indices in the product over $\tau'$ — this shows that the word $a'_{i_2, \ldots, i_\ell, b} \mathbf{W}_{i_1} \mathbf{W}_{i_2} \cdots \mathbf{W}_{i_\ell} \mathbf{W}_b$ appearing in $\mathbf{W}_{i_1} \Pi_{i_1}$ is equal to

$$\mathbf{W}_{i_1} \left( \frac{1}{B!} \sum_{\tau' \in \mathcal{T}'} (-\alpha)^\ell \mathbf{W}_{i_2} \cdots \mathbf{W}_{i_\ell} \mathbf{W}_b \right) = a_{i_1, \ldots, i_\ell} \mathbf{W}_{i_1} \mathbf{W}_{i_2} \cdots \mathbf{W}_{i_\ell} \mathbf{W}_b.$$

Thus, we conclude that $\sum_{b=1}^{B} \Pi_b \mathbf{W}_b = \sum_{b=1}^{B} \mathbf{W}_b \Pi_b$, and hence $\mathbf{Z}$ is symmetric.

Next, we will prove that $\mathrm{Range}(\mathbf{Z}) \subseteq \mathrm{Range}(\widetilde{\mathbf{X}}^\mathsf{T}) \subseteq \mathrm{Range}(\mathbf{X}^\mathsf{T})$. Since $\mathbf{Z}\mathbf{w} = \frac{1}{n} \sum_{b=1}^{B} \widetilde{\mathbf{X}}_b^\mathsf{T} \mathbf{X}_b \mathbf{w}$ for any $\mathbf{w} \in \mathbb{R}^p$, it is clear that $\mathrm{Range}(\mathbf{Z}) \subseteq \mathrm{Range}(\widetilde{\mathbf{X}}^\mathsf{T})$. Next, let $\mathbf{y} \in \mathbb{R}^n$ be a generic vector partitioned into $\mathbf{y}_1, \ldots, \mathbf{y}_B$ in the same way as the batches $\mathbf{X}_1, \ldots, \mathbf{X}_b$. From expanding the product in $\Pi_b$, we can write $\widetilde{\mathbf{X}}^\mathsf{T} \mathbf{y} = \sum_{b=1}^{B} \Pi_b \mathbf{X}_b^\mathsf{T} \mathbf{y}_b = \sum_{b=1}^{B} \mathbf{X}_b^\mathsf{T} \mathbf{y}_b + \sum_{b=1}^{B} \alpha_b \mathbf{X}_b^\mathsf{T} \mathbf{X}_b \mathbf{v}_b$ for some coefficients $\alpha_b \in \mathbb{R}$ and vectors $\mathbf{v}_b$. Hence, $\mathrm{Range}(\widetilde{\mathbf{X}}^\mathsf{T}) \subseteq \mathrm{Range}(\mathbf{X}^\mathsf{T})$. ∎

### B.1.2. Proof of Theorem 2

**Lemma 17** *Let $\mathbf{A} \in \mathbb{R}^{p \times p}$ be a symmetric matrix, and define $\mathbf{P} = (\mathbf{I} - \mathbf{A})(\mathbf{I} - \mathbf{A})^\dagger$ and $\mathbf{P}_0 = \mathbf{I} - \mathbf{P}$ to be the orthogonal projectors onto the range and nullspace of $\mathbf{I} - \mathbf{A}$ respectively. Then we have*

$$(\mathbf{I} + \mathbf{A} + \cdots + \mathbf{A}^{k-1}) = (\mathbf{I} - \mathbf{A}^k)(\mathbf{I} - \mathbf{A})^\dagger + k\mathbf{P}_0.$$

**Proof** Since $\mathbf{I} = \mathbf{P} + \mathbf{P}_0$, we can write $(\mathbf{I} + \mathbf{A} + \cdots + \mathbf{A}^{k-1}) = (\mathbf{I} + \mathbf{A} + \cdots + \mathbf{A}^{k-1})(\mathbf{P} + \mathbf{P}_0)$. By multiplying both sides of the algebraic identity $(\mathbf{I} + \mathbf{A} + \cdots + \mathbf{A}^{k-1})(\mathbf{I} - \mathbf{A}) = (\mathbf{I} - \mathbf{A}^k)$ by $(\mathbf{I} - \mathbf{A})^\dagger$, we have $(\mathbf{I} + \mathbf{A} + \cdots + \mathbf{A}^{k-1})\mathbf{P} = (\mathbf{I} - \mathbf{A}^k)(\mathbf{I} - \mathbf{A})^\dagger$, which yields the first term. For the second term, note that $\mathbf{A}^\ell \mathbf{P}_0 = \mathbf{P}_0$ for any $\ell \geq 1$, since $\mathbf{A}\mathbf{x} = \mathbf{x}$ for any $\mathbf{x}$ in the nullspace of $\mathbf{I} - \mathbf{A}$. Thus, $(\mathbf{I} + \mathbf{A} + \cdots + \mathbf{A}^{k-1})\mathbf{P}_0 = k\mathbf{P}_0$, which yields the second term. ∎

**Proof** [Proof of Theorem 2] Recall that from (2.2), the iterates $\boldsymbol{\beta}_k^{(b)}$ from mini-batch gradient descent after $b$ iterations over the mini-batches in the $k$th epoch satisfy

$$\boldsymbol{\beta}_k^{(b)} = \boldsymbol{\beta}_k^{(b)} - \frac{B\alpha}{n} \mathbf{X}_{\tau(b)}^\mathsf{T} (\mathbf{X}_{\tau(b)} \boldsymbol{\beta}_k^{(b-1)} - \mathbf{y}_{\tau(b)}), \quad b = 1, 2, \ldots, B,$$

given a permutation $\tau = (\tau(1), \tau(2), \ldots, \tau(B))$ of the mini-batches in the $k$th epoch, where $\boldsymbol{\beta}_k^{(0)} := \boldsymbol{\beta}_{k-1}^{(B)}$ and $\boldsymbol{\beta}_0^{(B)} := \boldsymbol{\beta}_0$. By using the fact that $\mathbf{y}_b = \mathbf{X}_b \boldsymbol{\beta}_* + \boldsymbol{\eta}_b$ for each mini-batch, the displayed equation above rearranges to

$$\boldsymbol{\beta}_k^{(b)} - \boldsymbol{\beta}_* = \left( \mathbf{I} - \frac{B\alpha}{n} \mathbf{X}_{\tau(b)}^\mathsf{T} \mathbf{X}_{\tau(b)} \right) (\boldsymbol{\beta}_k^{(b-1)} - \boldsymbol{\beta}_*) + \frac{B\alpha}{n} \mathbf{X}_{\tau(b)}^\mathsf{T} \boldsymbol{\eta}_{\tau(b)}, \quad b = 1, 2, \ldots, B.$$

By iterating this relationship, we deduce that the estimate at the end of the $k$th epoch satisfies

$$\boldsymbol{\beta}_k^{(B)} - \boldsymbol{\beta}_* = \prod_{b=1}^{B} (\mathbf{I} - \alpha \mathbf{W}_{\tau(b)})(\boldsymbol{\beta}_{k-1}^{(B)} - \boldsymbol{\beta}_*) + \frac{B\alpha}{n} \sum_{b=1}^{B} \prod_{j:j > \tau^{-1}(b)} (\mathbf{I} - \alpha \mathbf{W}_{\tau(j)}) \mathbf{X}_b^\mathsf{T} \boldsymbol{\eta}_b. \quad \text{(B.4)}$$

Recall that $\bar{\boldsymbol{\beta}}_k = \mathbb{E}_{\tau \sim \mathrm{Unif}(S_B)}\left[\boldsymbol{\beta}_k^{(B)}\right]$. Hence, by taking the expectation over the random permutations of the batches in each epoch, drawn uniformly from the $B!$ permutations in the symmetric group $S_B$ of $B$ elements, the error vector $\bar{\boldsymbol{\beta}}_k - \boldsymbol{\beta}_*$ satisfies the recursive relationship

$$
\begin{aligned}
\bar{\boldsymbol{\beta}}_k - \boldsymbol{\beta}_* = {} & \frac{1}{B!} \sum_{\tau \in S_B} \prod_{b=1}^{B} (\mathbf{I} - \alpha \mathbf{W}_{\tau(b)})(\bar{\boldsymbol{\beta}}_{k-1} - \boldsymbol{\beta}_*) \\
& + \frac{B\alpha}{n} \left\{ \frac{1}{B!} \sum_{\tau \in S_B} \sum_{b=1}^{B} \prod_{j:j>\tau^{-1}(b)} (\mathbf{I} - \alpha \mathbf{W}_{\tau(j)}) \mathbf{X}_b^{\mathsf{T}} \right\} \boldsymbol{\eta}_b.
\end{aligned}
\tag{B.5}
$$

By moving the sum over $b$ outside, recognizing the definition of $\Pi_b$ from (B.2), and recalling that $\widetilde{\mathbf{X}}_b^{\mathsf{T}} = \Pi_b \mathbf{X}_b$, the second term is equal to

$$
\frac{B\alpha}{n} \sum_{b=1}^{B} \widetilde{\mathbf{X}}_b^{\mathsf{T}} \boldsymbol{\eta}_b = \frac{B\alpha}{n} \widetilde{\mathbf{X}}^{\mathsf{T}} \boldsymbol{\eta}.
$$

Next, by writing $\mathbf{Z} = \frac{1}{n} \sum_{b=1}^{B} \widetilde{\mathbf{X}}_b^{\mathsf{T}} \mathbf{X}_b$, we have

$$
\mathbf{Z} = \frac{1}{B\alpha} \left( \frac{1}{B!} \sum_{\tau \in S_B} \sum_{b=1}^{B} \prod_{j:j>\tau^{-1}(b)} (\mathbf{I} - \alpha \mathbf{W}_{\tau(j)}) \alpha \mathbf{W}_b \right).
\tag{B.6}
$$

We claim that the identity

$$
\frac{1}{B!} \sum_{\tau \in S_B} \sum_{b=1}^{B} \prod_{j:j>\tau^{-1}(b)} (\mathbf{I} - \alpha \mathbf{W}_{\tau(j)}) \alpha \mathbf{W}_b = \mathbf{I} - \frac{1}{B!} \sum_{\tau \in S_B} \prod_{b=1}^{B} (\mathbf{I} - \alpha \mathbf{W}_{\tau(b)})
\tag{B.7}
$$

holds. Assuming that this is true for now, combining (B.6) and (B.7) shows that (B.5) can be written as

$$
\bar{\boldsymbol{\beta}}_k - \boldsymbol{\beta}_* = (\mathbf{I} - B\alpha \mathbf{Z})(\bar{\boldsymbol{\beta}}_{k-1} - \boldsymbol{\beta}_*) + \frac{B\alpha}{n} \widetilde{\mathbf{X}}^{\mathsf{T}} \boldsymbol{\eta}.
\tag{B.8}
$$

Hence, by recursively applying this relationship, we obtain

$$
\bar{\boldsymbol{\beta}}_k - \boldsymbol{\beta}_* = (\mathbf{I} - B\alpha \mathbf{Z})^k (\boldsymbol{\beta}_0 - \boldsymbol{\beta}_*) + \frac{B\alpha}{n} \sum_{j=1}^{k} (\mathbf{I} - B\alpha \mathbf{Z})^{k-j} \widetilde{\mathbf{X}}^{\mathsf{T}} \boldsymbol{\eta}.
$$

The proof of (3.6) is completed by using the following identity from Lemma 17 to simplify the expression for the sum above:

$$
\sum_{j=1}^{k} (\mathbf{I} - B\alpha \mathbf{Z})^{k-j} \widetilde{\mathbf{X}}^{\mathsf{T}} = \left[ \mathbf{I} - (\mathbf{I} - B\alpha \mathbf{Z})^k \right] (B\alpha \mathbf{Z})^{\dagger} \widetilde{\mathbf{X}}^{\mathsf{T}}.
$$

Here, we use the assumption that $\mathrm{Range}(\widetilde{\mathbf{X}}^{\mathsf{T}}) \subseteq \mathrm{Range}(\widetilde{\mathbf{X}}^{\mathsf{T}} \mathbf{X}) = \mathrm{Range}(\mathbf{Z})$ to deduce that $\mathbf{P}_{\mathbf{Z},0} \widetilde{\mathbf{X}}^{\mathsf{T}} = \mathbf{0}$. Furthermore, by incorporating the decomposition of the initial error

$$
\boldsymbol{\beta}_0 - \boldsymbol{\beta}_* = \mathbf{P}_{\mathbf{Z},0}(\boldsymbol{\beta}_0 - \boldsymbol{\beta}_*) + \mathbf{P}_{\mathbf{Z}}(\boldsymbol{\beta}_0 - \boldsymbol{\beta}_*),
$$

noting that $(\mathbf{I} - B\alpha\mathbf{Z})^k \mathbf{P}_{\mathbf{Z},0} = \mathbf{P}_{\mathbf{Z},0}$, we obtain (3.7).

Finally, it remains to prove that the identity (B.7) holds. We prove the equivalent identity, noting that $|S_B| = B!$ so that the identity matrix $\mathbf{I}$ can be brought inside the sum:

$$\frac{1}{B!} \sum_{\tau \in S_B} \sum_{b=1}^{B} \prod_{j:j>\tau^{-1}(b)} (\mathbf{I} - \alpha\mathbf{W}_{\tau(j)})\alpha\mathbf{W}_b = \frac{1}{B!} \sum_{\tau \in S_B} \left( \mathbf{I} - \prod_{b=1}^{B} (\mathbf{I} - \alpha\mathbf{W}_{\tau(b)}) \right). \tag{B.9}$$

We will prove this by matching each summand on the left hand side to a summand on the right hand side. Fix a permutation $\tau \in S_B$; without loss of generality, we may assume that $\tau = (1, 2, \dots, B - 1, B)$. On the left hand side, the summand corresponding to $\tau$ is

$$\alpha\mathbf{W}_B + (\mathbf{I} - \alpha\mathbf{W}_B)\alpha\mathbf{W}_{B-1} + \cdots + (\mathbf{I} - \alpha\mathbf{W}_B)\cdots(\mathbf{I} - \alpha\mathbf{W}_3)(\mathbf{I} - \alpha\mathbf{W}_2)\alpha\mathbf{W}_1. \tag{B.10}$$

On the right hand side, the summand corresponding to $\tau$ is

$$\mathbf{I} - (\mathbf{I} - \alpha\mathbf{W}_B)(\mathbf{I} - \alpha\mathbf{W}_{B-1})\cdots(\mathbf{I} - \alpha\mathbf{W}_2)(\mathbf{I} - \alpha\mathbf{W}_1). \tag{B.11}$$

Consider expanding the product by choosing a term from each bracket going from right to left. For the last bracket, choosing $\alpha\mathbf{W}_1$ yields the term $(\mathbf{I} - \alpha\mathbf{W}_B)\cdots(\mathbf{I} - \alpha\mathbf{W}_3)(\mathbf{I} - \alpha\mathbf{W}_2)\alpha\mathbf{W}_1$ ending in $\alpha\mathbf{W}_1$, matching the left hand side. Otherwise, choosing $\mathbf{I}$ results in a smaller product to which the same argument can be applied recursively. In the end, we are left with the single term $(\alpha\mathbf{W}_B - \mathbf{I}) - \mathbf{I}$, so that the identity vanishes and we are left with $\alpha\mathbf{W}_B$. Thus, we see that (B.10) and (B.11) correspond to the exact same expression, and summing over all $\tau \in S_B$ completes the proof of the claim (B.9). ∎

**Proof** [Proof of Corollary 4] Recall that $\mathbf{P}_{\mathbf{Z},0} = \mathbf{I} - \mathbf{Z}^\dagger\mathbf{Z}$ and $\mathbf{Z} = \frac{1}{n}\widetilde{\mathbf{X}}^\mathsf{T}\mathbf{X}$. From (3.6) and (3.7) of Theorem 2, it is clear that if $\|(\mathbf{I} - B\alpha\mathbf{W})\mathbf{P}_{\mathbf{Z}}\| < 1$, then $\bar{\beta}_k$ converges as $k \to \infty$ to the vector

$$\mathbf{P}_{\mathbf{Z},0}\boldsymbol{\beta}_0 + \mathbf{Z}^\dagger\mathbf{Z}\boldsymbol{\beta}_* + \frac{1}{n}\mathbf{Z}^\dagger\widetilde{\mathbf{X}}^\mathsf{T}\boldsymbol{\eta} = \mathbf{P}_{\mathbf{Z},0}\boldsymbol{\beta}_0 + (\widetilde{\mathbf{X}}^\mathsf{T}\mathbf{X})^\dagger\widetilde{\mathbf{X}}^\mathsf{T}(\mathbf{X}\boldsymbol{\beta}_* + \boldsymbol{\eta}).$$

Since $\mathbf{y} = \mathbf{X}\boldsymbol{\beta}_* + \boldsymbol{\eta}$, we obtain the claimed expression for the limiting vector $\bar{\beta}_\infty$. ∎

### B.1.3. ON THE ASSUMPTIONS IN THEOREM 2

In this section, we expand upon the discussion on the assumption in Theorem 2 that $\mathrm{Range}(\widetilde{\mathbf{X}}^\mathsf{T}) \subseteq \mathrm{Range}(\widetilde{\mathbf{X}}^\mathsf{T}\mathbf{X})$.

- In the overparameterized case ($p \geq n$), this follows if $\mathbf{X} \in \mathbb{R}^{n \times p}$ has rank $n$. Thus, for any $\boldsymbol{\eta} \in \mathbb{R}^n$, we can write $\boldsymbol{\eta} = \mathbf{X}\boldsymbol{\theta}$ for some $\boldsymbol{\theta} \in \mathbb{R}^p$. Hence, $\widetilde{\mathbf{X}}^\mathsf{T}\boldsymbol{\eta} = \widetilde{\mathbf{X}}^\mathsf{T}\mathbf{X}\boldsymbol{\theta} \in \mathrm{Range}(\widetilde{\mathbf{X}}^\mathsf{T}\mathbf{X})$.

- In the underparameterized case ($p < n$), this also follows if we assume that $\widetilde{\mathbf{X}}^\mathsf{T}\mathbf{X}$ (or equivalently $\mathbf{Z}$) has rank $p$ since $\mathrm{Range}(\widetilde{\mathbf{X}}^\mathsf{T}\mathbf{X}) = \mathbb{R}^p$.

  Next, using less trivial assumptions, the condition also follows if we assume that $\mathrm{Range}(\widetilde{\mathbf{X}}) \subseteq \mathrm{Range}(\mathbf{X})$. For $\boldsymbol{\eta} \in \mathbb{R}^n$, let $\widetilde{\mathbf{X}}^\mathsf{T}\mathbf{X}\boldsymbol{\theta}$ be the projection of $\widetilde{\mathbf{X}}^\mathsf{T}\boldsymbol{\eta}$ onto $\mathrm{Range}(\widetilde{\mathbf{X}}^\mathsf{T}\mathbf{X})$, where $\boldsymbol{\theta} \in \mathbb{R}^p$. Thus, $\widetilde{\mathbf{X}}^\mathsf{T}\boldsymbol{\eta} - \widetilde{\mathbf{X}}^\mathsf{T}\mathbf{X}\boldsymbol{\theta}$ is orthogonal to $\mathrm{Range}(\widetilde{\mathbf{X}}^\mathsf{T}\mathbf{X})$, or in other words,

$$\mathbf{X}^\mathsf{T}\widetilde{\mathbf{X}}\widetilde{\mathbf{X}}^\mathsf{T}(\boldsymbol{\eta} - \mathbf{X}\boldsymbol{\theta}) = \mathbf{0}.$$

We claim that $\widetilde{\mathbf{X}}^\mathsf{T}\boldsymbol{\eta} = \widetilde{\mathbf{X}}^\mathsf{T}\mathbf{X}\boldsymbol{\theta}$. Since $\mathrm{Range}(\widetilde{\mathbf{X}}) \subseteq \mathrm{Range}(\mathbf{X})$, we have $\widetilde{\mathbf{X}}\widetilde{\mathbf{X}}^\mathsf{T}(\boldsymbol{\eta} - \mathbf{X}\boldsymbol{\theta}) \in \mathrm{Range}(\mathbf{X})$, and thus $\mathbf{X}^\mathsf{T}\widetilde{\mathbf{X}}\widetilde{\mathbf{X}}^\mathsf{T}(\boldsymbol{\eta} - \mathbf{X}\boldsymbol{\theta}) = \mathbf{0}$ if and only if $\widetilde{\mathbf{X}}\widetilde{\mathbf{X}}^\mathsf{T}(\boldsymbol{\eta} - \mathbf{X}\boldsymbol{\theta}) = \mathbf{0}$. Furthermore, $\widetilde{\mathbf{X}}\widetilde{\mathbf{X}}^\mathsf{T}(\boldsymbol{\eta} - \mathbf{X}\boldsymbol{\theta}) = \mathbf{0}$ if and only if $\widetilde{\mathbf{X}}^\mathsf{T}(\boldsymbol{\eta} - \mathbf{X}\boldsymbol{\theta}) = \mathbf{0}$, which completes the proof.

The assumption in the overparameterized case (which is arguably the more interesting case for machine learning applications) is natural, and does not depend on the structure of the mini-batches or the step size. The underparameterized case seems to be more delicate, and it remains unclear what the necessary assumptions on the structure of the mini-batches or on the step size are in this regime for the required condition to hold. However, in our numerical experiments, $\widetilde{\mathbf{X}}^\mathsf{T}\mathbf{X}$ was always observed to have the same rank as $\mathbf{X}$, so the assumption is likely to be typically satisfied generically.

In fact, we observed that $\mathrm{Range}(\mathbf{X}^\mathsf{T})$ and $\mathrm{Range}(\widetilde{\mathbf{X}}^\mathsf{T}\mathbf{X})$ always appeared to be very similar, if not identical, which suggests that it may be possible to prove that the two subspaces coincide under a set of generic assumptions.

### B.1.4. MINI-BATCHING WITH REPLACEMENT

Here, we will provide more details on our claims in Remark 3 on the error dynamics when the mini-batches are sampled *with replacement*. Specifically, suppose that in each iteration, we sample a mini-batch with replacement uniformly at random from the same set of $B$ mini-batches $\mathbf{X}_1, \ldots, \mathbf{X}_b$ instead. If $\widehat{\boldsymbol{\beta}}_k$ denotes the mean iterate after $k$ epochs (which corresponds to $Bk$ iterations), averaged over the i.i.d. sampling of the mini-batches in each iteration, then we will show that

$$\widehat{\boldsymbol{\beta}}_k - \boldsymbol{\beta}_* = (\mathbf{I} - \alpha\mathbf{W})^{Bk}(\boldsymbol{\beta}_0 - \boldsymbol{\beta}_*) + \frac{1}{n}\left[\mathbf{I} - (\mathbf{I} - \alpha\mathbf{W})^{Bk}\right]\mathbf{W}^\dagger\mathbf{X}^\mathsf{T}\boldsymbol{\eta}. \tag{B.12}$$

Thus, we see that the error dynamics when sampling with replacement are essentially identical to those of full-batch gradient descent up to a time change by a factor of $B$.

**Proof** [Proof of (B.12)] Let $\widehat{\boldsymbol{\beta}}_{k,t}$ denote the mean parameters in the $k$th epoch after $t$ iterations. (Thus, $\widehat{\boldsymbol{\beta}}_k = \widehat{\boldsymbol{\beta}}_{k,0} = \widehat{\boldsymbol{\beta}}_{k-1,B}$.) Note that $\frac{1}{n}\sum_{b=1}^{B}\mathbf{X}_b^\mathsf{T}\mathbf{X}_b = \frac{1}{n}\mathbf{X}^\mathsf{T}\mathbf{X} = \mathbf{W}$, and $\sum_{b=1}^{B}\mathbf{X}_b^\mathsf{T}\boldsymbol{\eta}_b = \mathbf{X}^\mathsf{T}\boldsymbol{\eta}$. Conditional on the iterate $\widehat{\boldsymbol{\beta}}_{k,t}$, the next mini-batch is sampled uniformly at random from the $B$ mini-batches $\mathbf{X}_1, \ldots \mathbf{X}_B$. Hence, we can develop the following recursive expression for the expected error vector after one iteration:

$$\widehat{\boldsymbol{\beta}}_{k,t+1} - \boldsymbol{\beta}_* = \frac{1}{B}\sum_{b=1}^{B}\left\{\left(\mathbf{I} - \frac{B\alpha}{n}\mathbf{X}_b^\mathsf{T}\mathbf{X}_b\right)(\widehat{\boldsymbol{\beta}}_{k,t} - \boldsymbol{\beta}_*) - \frac{B\alpha}{n}\mathbf{X}_b^\mathsf{T}\boldsymbol{\eta}_b\right\}$$

$$= (\mathbf{I} - \alpha\mathbf{W})(\widehat{\boldsymbol{\beta}}_{k,t} - \boldsymbol{\beta}_*) - \frac{\alpha}{n}\mathbf{X}^\mathsf{T}\boldsymbol{\eta}.$$

By iterating over $Bk$ iterations until the end of the $k$th epoch, and simplifying the matrix geometric series using Lemma 17, we obtain (B.12). ∎

### B.1.5. PROOF OF THEOREM 8

By expanding the square in (3.6) of Theorem 2 and using the fact that the cross-terms vanish upon taking expectation with respect to the mean-zero noise $\boldsymbol{\eta}$, denoted by $\mathbb{E}_{\boldsymbol{\eta}}$, the generalization error $R_\mathbf{X}(\bar{\boldsymbol{\beta}}_k)$ is equal to

$$\mathbb{E}_{\boldsymbol{\eta}}\|\bar{\boldsymbol{\beta}}_k - \boldsymbol{\beta}_*\|_\Sigma^2 = \|\Sigma^{1/2}(\mathbf{I} - B\alpha\mathbf{Z})^k(\boldsymbol{\beta}_0 - \boldsymbol{\beta}_*)\|_2^2 + \mathbb{E}_{\boldsymbol{\eta}}\left\|\frac{1}{n}\Sigma^{1/2}\left[\mathbf{I} - (\mathbf{I} - B\alpha\mathbf{Z})^k\right]\mathbf{Z}^\dagger\widetilde{\mathbf{X}}^\mathsf{T}\boldsymbol{\eta}\right\|_2^2.$$

Since $\mathbf{Z}$ is symmetric, the first term is equal to $(\boldsymbol{\beta}_0 - \boldsymbol{\beta}_*)^{\mathsf{T}}(\mathbf{I} - B\alpha\mathbf{Z})^k\Sigma(\mathbf{I} - B\alpha\mathbf{Z})^k(\boldsymbol{\beta}_0 - \boldsymbol{\beta}_*)$. When combined with the decomposition of the initial error in (3.7), this yields the first two terms of the claimed generalization error, corresponding to the bias. The second term of the expansion above, written as a trace using the cyclic property, is equal to

$$\frac{1}{n^2}\operatorname{Tr}\left(\Sigma\left[\mathbf{I} - (\mathbf{I} - B\alpha\mathbf{Z})^k\right]\mathbf{Z}^\dagger\widetilde{\mathbf{X}}^{\mathsf{T}}\mathbb{E}_{\boldsymbol{\eta}}[\boldsymbol{\eta}\boldsymbol{\eta}^{\mathsf{T}}]\widetilde{\mathbf{X}}\mathbf{Z}^\dagger\left[\mathbf{I} - (\mathbf{I} - B\alpha\mathbf{Z})^k\right]\right).$$

Since $\mathbb{E}_{\boldsymbol{\eta}}[\boldsymbol{\eta}\boldsymbol{\eta}^{\mathsf{T}}] = \sigma^2\mathbf{I}$, this completes the proof. ∎

## B.2. Two-batch gradient descent

For the following proofs, we use the Loewner order defined by the cone of positive semidefinite matrices: that is, for symmetric matrices $\mathbf{A}, \mathbf{B}$, we have $\mathbf{A} \preceq \mathbf{B}$ if and only if $\mathbf{B} - \mathbf{A}$ is positive semidefinite, or equivalently $\mathbf{x}^{\mathsf{T}}\mathbf{A}\mathbf{x} \leq \mathbf{x}^{\mathsf{T}}\mathbf{B}\mathbf{x}$ for all unit vectors $\mathbf{x}$. We recall some basic properties of the Loewner order: if $\mathbf{A} \preceq \mathbf{B}$ and $\mathbf{C} \preceq \mathbf{D}$, then

- (Preserved by conjugation) $\mathbf{C}^{\mathsf{T}}\mathbf{A}\mathbf{C} \preceq \mathbf{C}^{\mathsf{T}}\mathbf{B}\mathbf{C}$ for any $\mathbf{C}$ with compatible dimensions

- $\mathbf{A} + \mathbf{B} \preceq \mathbf{C} + \mathbf{D}$ and $\alpha\mathbf{A} \preceq \alpha\mathbf{B}$ for any $\alpha \geq 0$.

- (Preserved by trace) $\operatorname{Tr}\mathbf{A} \leq \operatorname{Tr}\mathbf{B}$.

Furthermore, recall that $\mathbf{W}_1 + \mathbf{W}_2 = 2n^{-1}\mathbf{X}^{\mathsf{T}}\mathbf{X}$. Therefore, the assumption $\alpha \leq 1/(n^{-1}\|\mathbf{X}^{\mathsf{T}}\mathbf{X}\|)$ is simply the same as $\alpha \leq 2/\|\mathbf{W}_1 + \mathbf{W}_2\|$ in different notation.

### B.2.1. PROOF OF PROPOSITION 7

The claim follows if we can show that $\mathbf{Z} \succ \mathbf{0}$ and $2\alpha\mathbf{Z} \prec 2\mathbf{I}$, assuming $\alpha\|\mathbf{W}_1 + \mathbf{W}_2\| < 2$.

- $\mathbf{Z} \succ \mathbf{0}$:[9] the key observation is that we can write

$$\mathbf{Z} = \frac{1}{2}(\mathbf{W}_1 + \mathbf{W}_2) - \frac{1}{4}\alpha(\mathbf{W}_1 + \mathbf{W}_2)^2 + \frac{1}{4}\alpha(\mathbf{W}_1^2 + \mathbf{W}_2^2)$$
$$= \frac{1}{2}(\mathbf{W}_1 + \mathbf{W}_2)\left[\mathbf{I} - \frac{1}{2}\alpha(\mathbf{W}_1 + \mathbf{W}_2)\right] + \frac{1}{4}\alpha(\mathbf{W}_1^2 + \mathbf{W}_2^2).$$

Since $\mathbf{W}_1, \mathbf{W}_2 \succeq \mathbf{0}$, we have $\mathbf{W}_1^2 + \mathbf{W}_2^2 \succeq \mathbf{0}$, and using the assumption $\frac{1}{2}\alpha(\mathbf{W}_1 + \mathbf{W}_2) \prec \mathbf{I}$, we deduce that the first term is also positive semidefinite. Hence, $\mathbf{Z} \succeq \mathbf{0}$.

- $2\alpha\mathbf{Z} \prec 2\mathbf{I}$: we can write

$$2\alpha\mathbf{Z} = \alpha\mathbf{W}_1\left(\mathbf{I} - \frac{1}{2}\alpha\mathbf{W}_2\right) + \alpha\mathbf{W}_2\left(\mathbf{I} - \frac{1}{2}\alpha\mathbf{W}_1\right)$$
$$= \alpha\left(\mathbf{W}_1 + \mathbf{W}_2\right)\left(2\mathbf{I} - \frac{1}{2}\alpha(\mathbf{W}_1 + \mathbf{W}_2)\right) - \alpha\mathbf{W}_1\left(\mathbf{I} - \frac{1}{2}\alpha\mathbf{W}_1\right) - \alpha\mathbf{W}_2\left(\mathbf{I} - \frac{1}{2}\alpha\mathbf{W}_2\right).$$

---

9. Even though $\mathbf{W}_1 + \mathbf{W}_2 \succeq \mathbf{0}$, this is not immediately obvious since the anticommutator $\mathbf{W}_1\mathbf{W}_2 + \mathbf{W}_2\mathbf{W}_1$ is not positive semidefinite in general.

Since $\alpha\mathbf{W}_1 \prec 2\mathbf{I}$ and $\alpha\mathbf{W}_2 \prec 2\mathbf{I}$ by assumption, we have $(\mathbf{I}-\frac{1}{2}\alpha\mathbf{W}_1) \succ \mathbf{0}$ and $(\mathbf{I}-\frac{1}{2}\alpha\mathbf{W}_2) \succ \mathbf{0}$. Thus,

$$2\alpha\mathbf{Z} \prec 2\alpha\left(\mathbf{W}_1 + \mathbf{W}_2\right) - \frac{1}{2}\alpha^2\left(\mathbf{W}_1 + \mathbf{W}_2\right)^2.$$

By considering the eigenvalues of $\alpha(\mathbf{W}_1 + \mathbf{W}_2)$, which satisfy $\|\alpha(\mathbf{W}_1 + \mathbf{W}_2)\| < 2$ by assumption, we deduce that the operator norm of the upper bound is at most 2. Hence, we conclude that $2\alpha\mathbf{Z} \prec 2\mathbf{I}$. ∎

### B.2.2. PROOF OF PROPOSITION 10

Our goal is to bound the generalization error given in Theorem 8 (with $B = 2$) by bounding the trace term (corresponding to the variance component). The key observation is that in the two-batch case, we have the explicit relationship between $\frac{1}{n}\widetilde{\mathbf{X}}^\mathsf{T}\widetilde{\mathbf{X}}$ and $\mathbf{Z}$:

$$\frac{1}{n}\widetilde{\mathbf{X}}^\mathsf{T}\widetilde{\mathbf{X}} = \mathbf{Z} + \frac{\alpha}{4}\left[\left(\frac{1}{2}\alpha\mathbf{W}_1 - \mathbf{I}\right)\mathbf{W}_2\mathbf{W}_1 + \left(\frac{1}{2}\alpha\mathbf{W}_2 - \mathbf{I}\right)\mathbf{W}_1\mathbf{W}_2\right]. \qquad \text{(B.13)}$$

By using the property $\mathbf{Z}^\dagger\mathbf{Z}\mathbf{Z}^\dagger = \mathbf{Z}^\dagger$ of the pseudoinverse, and the fact that the trace preserves the Loewner order, the claimed upper bound follows if we can show that

$$\frac{1}{4}\left[\left(\frac{1}{2}\alpha\mathbf{W}_1 - \mathbf{I}\right)\mathbf{W}_2\mathbf{W}_1 + \left(\frac{1}{2}\alpha\mathbf{W}_2 - \mathbf{I}\right)\mathbf{W}_1\mathbf{W}_2\right] \preceq \frac{1}{2}\|\mathbf{W}_1 + \mathbf{W}_2\|\mathbf{Z}, \qquad \text{(B.14)}$$

assuming that $\alpha\|\mathbf{W}_1 + \mathbf{W}_2\| \leq 2$. Since $\mathbf{Z} = \frac{1}{2}(\mathbf{W}_1 + \mathbf{W}_2) - \frac{1}{4}\alpha(\mathbf{W}_2\mathbf{W}_1 + \mathbf{W}_1\mathbf{W}_2)$, the claim (B.14) is equivalent to showing that

$$\frac{\alpha}{8}\left(\mathbf{W}_2\mathbf{W}_1\mathbf{W}_2 + \mathbf{W}_1\mathbf{W}_2\mathbf{W}_1\right) - \frac{1}{4}(\mathbf{W}_2\mathbf{W}_1 + \mathbf{W}_1\mathbf{W}_2)$$
$$\preceq \frac{1}{4}\|\mathbf{W}_1 + \mathbf{W}_2\|(\mathbf{W}_1 + \mathbf{W}_2) - \frac{\alpha}{8}\|\mathbf{W}_1 + \mathbf{W}_2\|(\mathbf{W}_2\mathbf{W}_1 + \mathbf{W}_1\mathbf{W}_2),$$

or, by rearranging,

$$\frac{\alpha}{8}\left\{(\mathbf{W}_2\mathbf{W}_1\mathbf{W}_2 + \mathbf{W}_1\mathbf{W}_2\mathbf{W}_1) + \|\mathbf{W}_1 + \mathbf{W}_2\|(\mathbf{W}_2\mathbf{W}_1 + \mathbf{W}_1\mathbf{W}_2)\right\}$$
$$\preceq \frac{1}{4}\left\{\|\mathbf{W}_1 + \mathbf{W}_2\|(\mathbf{W}_1 + \mathbf{W}_2) + (\mathbf{W}_2\mathbf{W}_1 + \mathbf{W}_1\mathbf{W}_2)\right\} \qquad \text{(B.15)}$$

Since $\mathbf{W}_1 \preceq \|\mathbf{W}_1\|\mathbf{I} \preceq \|\mathbf{W}_1 + \mathbf{W}_2\|\mathbf{I}$, and similarly $\mathbf{W}_2 \preceq \|\mathbf{W}_1 + \mathbf{W}_2\|\mathbf{I}$, the left hand side of (B.15) is bounded from above in the Loewner order by

$$\frac{\alpha}{8}\|\mathbf{W}_1 + \mathbf{W}_2\|\left\{(\mathbf{W}_1^2 + \mathbf{W}_2^2) + (\mathbf{W}_2\mathbf{W}_1 + \mathbf{W}_1\mathbf{W}_2)\right\} \preceq \frac{1}{4}(\mathbf{W}_1 + \mathbf{W}_2)^2,$$

where we use the assumption $\alpha\|\mathbf{W}_1 + \mathbf{W}_2\| \leq 2$ for the second inequality. Next, since $\|\mathbf{W}_1 + \mathbf{W}_2\|(\mathbf{W}_1 + \mathbf{W}_2) \succeq \mathbf{W}_1^2 + \mathbf{W}_2^2$, the right hand side of (B.15) is bounded from below by

$$\frac{1}{4}\left\{(\mathbf{W}_1^2 + \mathbf{W}_2^2) + (\mathbf{W}_2\mathbf{W}_1 + \mathbf{W}_1\mathbf{W}_2)\right\} = \frac{1}{4}(\mathbf{W}_1 + \mathbf{W}_2)^2.$$

Combining the preceding two displayed equations shows that (B.15) holds. If $\mathbf{W}_1 = \mathbf{W}_2 = c\mathbf{I}$ and $\alpha = 2/c$ for some $c > 0$, then it is also clear that (B.15) holds with equality.

Similarly as above, the lower bound follows if we can show that

$$\frac{1}{4}\left[\left(\frac{1}{2}\alpha\mathbf{W}_1 - \mathbf{I}\right)\mathbf{W}_2\mathbf{W}_1 + \left(\frac{1}{2}\alpha\mathbf{W}_2 - \mathbf{I}\right)\mathbf{W}_1\mathbf{W}_2\right] \succeq -\frac{1}{2}\|\mathbf{W}_1 + \mathbf{W}_2\|\mathbf{Z}, \qquad \text{(B.16)}$$

assuming that $\alpha\|\mathbf{W}_1 + \mathbf{W}_2\| \leq 2$. This is equivalent to showing that

$$\frac{\alpha}{8}\left(\mathbf{W}_2\mathbf{W}_1\mathbf{W}_2 + \mathbf{W}_1\mathbf{W}_2\mathbf{W}_1\right) - \frac{1}{4}(\mathbf{W}_2\mathbf{W}_1 + \mathbf{W}_1\mathbf{W}_2)$$
$$\succeq -\frac{1}{4}\|\mathbf{W}_1 + \mathbf{W}_2\|(\mathbf{W}_1 + \mathbf{W}_2) + \frac{\alpha}{8}\|\mathbf{W}_1 + \mathbf{W}_2\|(\mathbf{W}_2\mathbf{W}_1 + \mathbf{W}_1\mathbf{W}_2).$$

By rearranging and using the fact that $\mathbf{W}_2\mathbf{W}_1\mathbf{W}_2 + \mathbf{W}_1\mathbf{W}_2\mathbf{W}_1 \succeq \mathbf{0}$, this is implied by

$$\frac{1}{4}\|\mathbf{W}_1 + \mathbf{W}_2\|(\mathbf{W}_1 + \mathbf{W}_2) \succeq \frac{1}{4}\left(\frac{\alpha}{2}\|\mathbf{W}_1 + \mathbf{W}_2\| + 1\right)(\mathbf{W}_2\mathbf{W}_1 + \mathbf{W}_1\mathbf{W}_2).$$

By using the assumption $\alpha\|\mathbf{W}_1 + \mathbf{W}_2\| \leq 2$, and the fact that $\|\mathbf{W}_1 + \mathbf{W}_2\|(\mathbf{W}_1 + \mathbf{W}_2) \succeq (\mathbf{W}_1 + \mathbf{W}_2)^2$, this is further implied by

$$\frac{1}{4}(\mathbf{W}_1 + \mathbf{W}_2)^2 \succeq \frac{1}{2}(\mathbf{W}_2\mathbf{W}_1 + \mathbf{W}_1\mathbf{W}_2).$$

Since $(\mathbf{W}_1 + \mathbf{W}_2)^2 = \mathbf{W}_1^2 + \mathbf{W}_2^2 + \mathbf{W}_2\mathbf{W}_1 + \mathbf{W}_1\mathbf{W}_2$, this is equivalent to

$$\frac{1}{4}(\mathbf{W}_1 - \mathbf{W}_2)^2 = \frac{1}{4}(\mathbf{W}_1^2 + \mathbf{W}_2^2 - \mathbf{W}_2\mathbf{W}_1 - \mathbf{W}_1\mathbf{W}_2) \succeq \mathbf{0},$$

which is indeed true, and hence we conclude that the claim (B.16) holds.

Finally, the convergence of $\bar{\beta}_k$ to $\bar{\beta}_\infty$ follows from Corollary 4 (with $B = 2$), using the sufficient condition on the step size $\alpha$ given in Proposition 7. The resulting bound for the limiting generalization error, expressed in terms of $\mathbf{Z}$, is obtained from the fact that $(\mathbf{I} - 2\alpha\mathbf{Z})^k \to 0$. ∎

## B.3. Asymptotic analysis

In this section, we consider the asymptotics of $\mathbf{Z}(\alpha/B)$ as $n \to \infty$ with $p$ fixed, and evaluate the impact on the error trajectory and generalization error of mini-batch gradient descent with random reshuffling as compared to full-batch gradient descent.

### B.3.1. PROOF OF PROPOSITION 12

As $n \to \infty$, each $\mathbf{W}_b$ tends to $\Sigma$ almost surely. Therefore, by independence, we deduce that on a set of probability one, $\mathbf{W}_b \to \Sigma$ for all $b = 1, \ldots, B$. Since $\mathbf{Z}(\alpha) = \frac{1}{B}\sum_{b=1}^{B}\mathbf{W}_b\Pi_b$, it suffices to compute the limiting expression for a fixed $\mathbf{W}_b\Pi_b$ by symmetry. The starting point is the expression for $\Pi_b$ from (B.3):

$$\Pi_b = \mathbf{I} - \sum_{i=1}^{B-1}\left\{\frac{(-1)^{i+1}}{(i+1)!} \cdot \alpha^i \sum_{\substack{\{b_1,\ldots,b_i\}\subseteq[B]\backslash b \\ b_1,\ldots,b_i \text{ distinct, ordered}}} \mathbf{W}_{b_1}\ldots\mathbf{W}_{b_i}\right\}.$$

Indeed, we simply have to count the number of terms for the internal summand for a fixed $i \in \{1, 2, \ldots, B-1\}$, which indicates the number of distinct sample covariances that appear. There are $i!\binom{B-1}{i}$ (ordered) ways to choose $i$ distinct indices $\{b_1, \ldots, b_i\}$ from $[B] \setminus \{b\}$. For each such choice, the limit of $\mathbf{W}_{b_1} \ldots \mathbf{W}_{b_i}$ is $\Sigma^i$. Therefore, introducing an extra factor of $\Sigma$ for $\mathbf{W}_b$ (which is not in any of the terms in $\Pi_b$), we have, as $n \to \infty$,

$$\mathbf{W}_b \Pi_b \to \Sigma - \sum_{i=1}^{B-1} (-1)^{i+1} \frac{\binom{B-1}{i}}{i+1} \alpha^i \Sigma^{i+1} = \Sigma \left\{ \mathbf{I} - \sum_{i=1}^{B-1} (-1)^{i+1} \frac{(B-1)!(B-1-i)!}{(i+1)!} \alpha^i \Sigma^i \right\}.$$

The proof is completed by replacing $\alpha$ with $\alpha/B$ to obtain the claimed expression for $\mathbf{Z}(\alpha/B)$. ∎

### B.3.2. PROOF OF PROPOSITION 13

Recall from Theorem 2 that the error trajectory of mini-batch gradient descent with random reshuffling and step size $\alpha/B$ is given by the following (with $\mathbf{Z} \equiv \mathbf{Z}(\alpha/B)$:

$$\bar{\boldsymbol{\beta}}_k - \boldsymbol{\beta}_* = (\mathbf{I} - \alpha\mathbf{Z})^k (\boldsymbol{\beta}_0 - \boldsymbol{\beta}_*) + \frac{1}{n} \left[ \mathbf{I} - (\mathbf{I} - \alpha\mathbf{Z})^k \right] \mathbf{Z}^\dagger \widetilde{\mathbf{X}}^\mathsf{T} \boldsymbol{\eta}.$$

By using the assumption that $\Pi_b \equiv \Pi = \mathbf{I} - p(\mathbf{W})$, we have

$$\mathbf{Z} = \frac{1}{n} \sum_{b=1}^{B} \Pi_b \mathbf{X}_b^\mathsf{T} \mathbf{X}_b = \Pi \left( \frac{1}{n} \sum_{b=1}^{B} \mathbf{X}_b^\mathsf{T} \mathbf{X}_b \right) = \Pi\mathbf{W}.$$

Since $\mathbf{Z}$ is symmetric, $\mathbf{Z} = \mathbf{W}\Pi = \mathbf{W}(\mathbf{I} - p(\mathbf{W}))$. Furthermore, $\widetilde{\mathbf{X}}^\mathsf{T} \boldsymbol{\eta} = \sum_{b=1}^{B} \Pi_b \mathbf{X}_b^\mathsf{T} \boldsymbol{\eta}_b = \Pi(\sum_{b=1}^{B} \mathbf{X}_b^\mathsf{T} \boldsymbol{\eta}_b) = \Pi\mathbf{X}^\mathsf{T}\boldsymbol{\eta}$. Therefore,

$$\mathbf{Z}^\dagger \widetilde{\mathbf{X}}^\mathsf{T} \boldsymbol{\eta} = \mathbf{W}^{-1}\Pi^{-1}\Pi\mathbf{X}^\mathsf{T}\boldsymbol{\eta} = \mathbf{W}^{-1}\mathbf{X}^\mathsf{T}\boldsymbol{\eta} = n\mathbf{X}^\dagger\boldsymbol{\eta}$$

Since $\mathbf{W} = \mathbf{V}\widehat{\mathbf{S}}\mathbf{V}^\mathsf{T}$, where $\mathbf{X} = \mathbf{U}\mathbf{S}\mathbf{V}^\mathsf{T}$ is the SVD of $\mathbf{X}$ with $\mathbf{U} \in \mathbb{R}^{n \times n}$ and $\mathbf{V} \in \mathbb{R}^{p \times p}$ orthogonal and $\mathbf{S} \in \mathbb{R}^{n \times p}$ diagonal, and $\widehat{\mathbf{S}} := \frac{1}{n}\mathbf{S}^\mathsf{T}\mathbf{S} \in \mathbb{R}^{p \times p}$, we have

$$\mathbf{V}^\mathsf{T}(\bar{\boldsymbol{\beta}}_k - \boldsymbol{\beta}_*) = [\mathbf{I} - \alpha\widehat{\mathbf{S}}(\mathbf{I} - p(\widehat{\mathbf{S}}))]^k \mathbf{V}^\mathsf{T}(\boldsymbol{\beta}_0 - \boldsymbol{\beta}_*) + \left[ \mathbf{I} - [\mathbf{I} - \alpha\widehat{\mathbf{S}}(\mathbf{I} - p(\widehat{\mathbf{S}}))]^k \right] \mathbf{S}^\dagger\mathbf{U}^\mathsf{T}\boldsymbol{\eta}.$$

The point of expressing the error in the eigenbasis $\mathbf{V}$ is that the dynamics decouple (since $\widehat{\mathbf{S}}$ is diagonal). Therefore, for $i = 1, 2, \ldots, p$, the $i$th coordinate of $\mathbf{V}^\mathsf{T}(\bar{\boldsymbol{\beta}}_k - \boldsymbol{\beta}_*)$ satisfies

$$[\mathbf{V}^\mathsf{T}(\bar{\boldsymbol{\beta}}_k - \boldsymbol{\beta}_*)]_i = [1 - \alpha\hat{\lambda}_i(1 - p(\hat{\lambda}_i))]^k [\mathbf{V}^\mathsf{T}(\boldsymbol{\beta}_0 - \boldsymbol{\beta}_*)]_i + \frac{1}{\hat{\lambda}_i} \left( 1 - [1 - \alpha\hat{\lambda}_i(1 - p(\hat{\lambda}_i))]^k \right) [\mathbf{U}^\mathsf{T}\boldsymbol{\eta}]_i.$$

Next, we can apply Theorem 8 for the corresponding generalization error:

$$R_{\mathbf{X}}(\bar{\boldsymbol{\beta}}_k) = (\boldsymbol{\beta}_0 - \boldsymbol{\beta}_*)^\mathsf{T}(\mathbf{I} - \alpha\mathbf{Z})^k \Sigma(\mathbf{I} - \alpha\mathbf{Z})^k(\boldsymbol{\beta}_0 - \boldsymbol{\beta}_*)$$
$$+ \frac{\sigma^2}{n} \operatorname{Tr} \left( \left[ \mathbf{I} - (\mathbf{I} - \alpha\mathbf{Z})^k \right] \Sigma \left[ \mathbf{I} - (\mathbf{I} - \alpha\mathbf{Z})^k \right] \mathbf{Z}^\dagger \left( \frac{1}{n}\widetilde{\mathbf{X}}^\mathsf{T}\widetilde{\mathbf{X}} \right) \mathbf{Z}^\dagger \right).$$

From the calculations above, we have $\mathbf{Z}^\dagger(\frac{1}{n}\widetilde{\mathbf{X}}^\mathsf{T}\widetilde{\mathbf{X}})\mathbf{Z}^\dagger = \mathbf{W}^{-1}(\frac{1}{n}\mathbf{X}^\mathsf{T}\mathbf{X})\mathbf{W}^{-1} = \mathbf{W}^{-1}$. Therefore, changing to the eigenbasis $\mathbf{V}$ again, we have

$$R_\mathbf{X}(\bar{\boldsymbol{\beta}}_k) = (\mathbf{V}^\mathsf{T}(\boldsymbol{\beta}_0 - \boldsymbol{\beta}_*))^\mathsf{T}[\mathbf{I} - \alpha\widehat{\mathbf{S}}(\mathbf{I} - p(\widehat{\mathbf{S}}))]^k \mathbf{V}^\mathsf{T}\Sigma\mathbf{V}[\mathbf{I} - \alpha\widehat{\mathbf{S}}(\mathbf{I} - p(\widehat{\mathbf{S}}))]^k(\mathbf{V}^\mathsf{T}(\boldsymbol{\beta}_0 - \boldsymbol{\beta}_*))$$
$$+ \frac{\sigma^2}{n}\operatorname{Tr}\left(\left[\mathbf{I} - [\mathbf{I} - \alpha\widehat{\mathbf{S}}(\mathbf{I} - p(\widehat{\mathbf{S}}))]^k\right]\mathbf{V}^\mathsf{T}\Sigma\mathbf{V}\left[\mathbf{I} - [\mathbf{I} - \alpha\widehat{\mathbf{S}}(\mathbf{I} - p(\widehat{\mathbf{S}}))]^k\right]\widehat{\mathbf{S}}^{-1}\right).$$

By using the assumption that $\mathbf{V}^\mathsf{T}\Sigma\mathbf{V} = \Lambda$ (i.e. that $\Sigma$ and $\mathbf{W}$ are simultaneously diagonalizable), then we have again obtained an expression that decouples since all the matrices involved are diagonal, and we can write the result as

$$R_\mathbf{X}(\bar{\boldsymbol{\beta}}_k) = \sum_{i=1}^p \lambda_i[1 - \alpha\hat{\lambda}_i(1 - p(\hat{\lambda}_i))]^{2k}[\mathbf{V}^\mathsf{T}(\boldsymbol{\beta}_0 - \boldsymbol{\beta}_*)]_i$$
$$+ \frac{\sigma^2}{n}\sum_{i=1}^p \frac{\lambda_i}{\hat{\lambda}_i}\left(1 - [1 - \alpha\hat{\lambda}_i(1 - p(\hat{\lambda}_i))]^k\right)^2.$$

This completes the proof of the claimed expressions for mini-batch gradient descent.

Finally, it is straightforward to show that the corresponding quantities for full-batch gradient descent (under the same assumptions) are the same as the given expressions with $\hat{\lambda}_i(1 - p(\hat{\lambda}_i))$ replaced by $\hat{\lambda}_i$ using the same strategy and the usual expressions for the error dynamics of full-batch gradient descent (Lemma 14 and Lemma 15). ∎

# Appendix C. Free probability computations

## C.1. Additional background

Techniques for computing the distribution of a sum or product of free random variables have been developed (e.g. see Mingo and Speicher, 2017; Raj Rao and Edelman, 2008). The reason that we could not apply these techniques is that while we could compute the distribution of $w_1 w_2$ or $w_2 w_1$ individually (where $w_1, w_2$ are free random variables), we cannot compute the distribution of $w_2 w_1 + w_1 w_2$ since the two summands are not freely independent.

More generally, the problem of describing the distribution of a *general polynomial of free random variables* in terms of its individual marginals—such as its density or smoothness properties—remains a difficult open problem, even in pure mathematics. Recent theoretical progress in Arizmendi et al. (2024) provides a general description of the atoms: in particular, Arizmendi et al. (2024, Theorem 1.3) implies that asymptotically, $\alpha\mathbf{Z}(\alpha/2)$ and $\mathbf{W}$ have the *same point mass* of $(1 - \gamma^{-1})$ at zero if and only if $\gamma > 1$ (i.e. in the overparameterized regime). To interpret this result, note that the point mass at zero effectively corresponds to the "dimension" of the frozen subspaces of weights for gradient descent; i.e. the rank of the projectors $\mathbf{P}_{\mathbf{Z},0}$ and $\mathbf{P}_0$ for mini-batch gradient descent with random reshuffling and full-batch gradient descent respectively.

## C.2. Algorithm

In this section, we describe our implementation of the general algorithm from Belinschi et al. (2017) for calculating the spectral distribution of the non-commutative polynomial

$$p(w_1, w_2) = w_1 + w_2 - \frac{1}{2}(w_2 w_1 + w_1 w_2)$$

of two freely independent Marchenko-Pastur distributions $w_1, w_2$ with ratio parameter $\gamma$ and variance $\alpha$. When $\gamma = 2\lim_{n,p\to\infty} p/n$, this corresponds to the limiting spectral distributions of the scaled sample covariances $\alpha\mathbf{W}_1$ and $\alpha\mathbf{W}_2$ of the two mini-batches in two-batch gradient descent with step size $\alpha$. For the statement of the algorithm for computing the spectral distributions of general polynomials of free random variables as well as the technical details and proofs, we refer to Belinschi et al. (2017) (and in particular, Theorem 4.1 and Theorem 2.2 of their paper).

First, we state some preliminaries on the Marchenko-Pastur distribution $\nu_{\gamma,\alpha}$ with ratio parameter $\gamma$ and variance $\alpha$. The *Stieltjes transform* of $\nu_{\gamma,\alpha}$ is given by

$$m_{\gamma,\alpha}(z) := \mathbb{E}_{Y\sim\nu_{\gamma,\alpha}}[(Y-z)^{-1}] = \frac{\alpha(1-\gamma) - z + \sqrt{(z-\alpha(\gamma+1))^2 - 4\gamma\alpha^2}}{2\alpha\gamma z} \tag{C.1}$$

for $z \in \mathbb{C}_+$, where $\mathbb{C}_+ = \{z \in \mathbb{C} : \text{Im}(z) > 0\}$ is the complex upper half-plane, and the branch of the complex square root is chosen with positive imaginary part. The *Cauchy transform* is given by $G(z) = -m(z)$. The Stieltjes transform of a real-valued random variable (or equivalently its Cauchy transform) uniquely determines its distribution through the Stieltjes inversion theorem (e.g. Mingo and Speicher, 2017, Theorem 6).

The algorithm of Belinschi et al. (2017) computes the Cauchy transform $G_p$ of $p(w_1, w_2)$, which uniquely determines its distribution, given the individual Cauchy transforms of $w_1, w_2$ by the following steps:

(1) Compute a *linearization* $\mathbf{L}_p(w_1, w_2)$ of the non-commutative polynomial $p(w_1, w_2) = w_1 + w_2 - \frac{1}{2}(w_2 w_1 + w_1 w_2)$ in the sense of (Belinschi et al., 2017, Definition 3.1): that is, we want to find

$$\mathbf{L}_p(w_1, w_2) = \begin{pmatrix} 0 & \mathbf{u}^{\mathsf{T}} \\ \mathbf{v} & \mathbf{Q} \end{pmatrix}$$

such that $p(w_1, w_2) = -\mathbf{u}^{\mathsf{T}}\mathbf{Q}^{-1}\mathbf{v}$, where $\mathbf{u}, \mathbf{v}$ are vectors with entries in $\mathbb{C}\langle w_1, w_2\rangle$, the algebra generated by $w_1, w_2$ over the field of complex numbers, and $\mathbf{Q}$ is a matrix with entries in $\mathbb{C}\langle w_1, w_2\rangle$. Specifically, we use

$$\mathbf{L}_p(w_1, w_2) = \begin{pmatrix} 0 & 1 & w_1 & w_2 \\ 1 & -1 & -1 & -1 \\ w_1 & -1 & -1 & 1 \\ w_2 & -1 & 1 & -1 \end{pmatrix}. \tag{C.2}$$

It may be easily checked that

$$\mathbf{Q}^{-1} = \begin{pmatrix} -1 & -1 & -1 \\ -1 & -1 & 1 \\ -1 & 1 & -1 \end{pmatrix}^{-1} = \frac{1}{2}\begin{pmatrix} 0 & -1 & -1 \\ -1 & 0 & 1 \\ -1 & 1 & 0 \end{pmatrix},$$

so that $w_1 + w_2 - \frac{1}{2}(w_2 w_1 + w_1 w_2) = -\mathbf{u}^{\mathsf{T}}\mathbf{Q}^{-1}\mathbf{v}$. We also define the matrices

$$\mathbf{b}_0 := \begin{pmatrix} 0 & 1 & 0 & 0 \\ 1 & -1 & -1 & -1 \\ 0 & -1 & -1 & 1 \\ 0 & -1 & 1 & -1 \end{pmatrix}, \quad \mathbf{b}_1 := \begin{pmatrix} 0 & 0 & 1 & 0 \\ 0 & 0 & 0 & 0 \\ 1 & 0 & 0 & 0 \\ 0 & 0 & 0 & 0 \end{pmatrix}, \quad \mathbf{b}_2 := \begin{pmatrix} 0 & 0 & 0 & 1 \\ 0 & 0 & 0 & 0 \\ 0 & 0 & 0 & 0 \\ 1 & 0 & 0 & 0 \end{pmatrix},$$

so that we can write $\mathbf{L}_p(w_1, w_2) = \mathbf{b}_0 \otimes 1 + \mathbf{b}_1 \otimes w_1 + \mathbf{b}_2 \otimes w_2$. Finally, $\mathbf{b}_1 \otimes w_1$ and $\mathbf{b}_2 \otimes w_2$ (i.e. matrices whose entries consist of $w_1$ and $w_2$ respectively) are freely independent operator-valued random variables.

(2) The *operator-valued Cauchy transform* $\mathbf{G}_{\mathbf{b}_1 \otimes w_1}(\mathbf{b})$ of $\mathbf{b}_1 \otimes w_1$ is defined by $\mathbf{G}_{\mathbf{b}_1 \otimes w_1}(\mathbf{b}) :=$ $\mathbb{E}\left[(\mathbf{b} - \mathbf{b}_1 \otimes w_1)^{-1}\right] = \int_{\mathbb{R}}(\mathbf{b} - t\mathbf{b}_j)^{-1}\mathrm{d}\nu_{\gamma,\alpha}(t)$ for complex-valued matrices $\mathbf{b}$ in the operator upper half-plane (i.e. whose imaginary part has only positive eigenvalues). By the Stieltjes inversion theorem, it can be calculated by the limiting formula

$$\mathbf{G}_{\mathbf{b}_1 \otimes w_1}(\mathbf{b}) = \lim_{\epsilon \downarrow 0} \frac{-1}{\pi} \int_{\mathbb{R}} (\mathbf{b} - t\mathbf{b}_1)^{-1}\mathrm{Im}(G_{w_1}(t + i\epsilon))\,\mathrm{d}t,$$

where the integral is taken elementwise, and the (scalar-valued) Cauchy transform $G_{w_1}$ for the distribution $\nu_{\gamma,\alpha}$ is (the negative) of (C.1) above. (In our implementation, we found that computing this integral with parameters $\epsilon \sim 10^{-6}$ and $t \sim 100$ worked well; in particular, $t$ does not need to be large since the matrices involved have bounded operator norm and the Marchenko-Pastur distribution has compact support.) Similarly, the operator-valued Cauchy transform $\mathbf{G}_{\mathbf{b}_2 \otimes w_2}$ of $\mathbf{b}_2 \otimes w_2$ can be computed in the same way with $\mathbf{b}_1, w_1$ replaced by $\mathbf{b}_2, w_2$.

(3) Let $f_{\mathbf{b}}$ be the map defined by

$$f_{\mathbf{b}}(\mathbf{a}) = \mathbf{h}_{\mathbf{b}_2 \otimes w_2}(\mathbf{h}_{\mathbf{b}_1 \otimes w_1}(\mathbf{a}) + \mathbf{b}) + \mathbf{b},$$

where $\mathbf{h}_{\mathbf{b}_1 \otimes w_1}(\mathbf{a}) = (\mathbf{G}_{\mathbf{b}_1 \otimes w_1}(\mathbf{a}))^{-1} - \mathbf{a}$ and $\mathbf{h}_{\mathbf{b}_2 \otimes w_2}(\mathbf{a}) = (\mathbf{G}_{\mathbf{b}_2 \otimes w_2}(\mathbf{a}))^{-1} - \mathbf{a}$ are the so-called "$h$-transforms" of $\mathbf{b}_1 \otimes w_1$ and $\mathbf{b}_2 \otimes w_2$ respectively.

The *operator-valued Cauchy transform of the sum* $\mathbf{b}_1 \otimes w_1 + \mathbf{b}_2 \otimes w_2$ satisfies $G_{\mathbf{b}_1 \otimes w_1 + \mathbf{b}_2 \otimes w_2}(\mathbf{b}) = G_{\mathbf{b}_1 \otimes w_1}(\omega(\mathbf{b}))$, where $\omega(\mathbf{b})$ is the *unique fixed point of the map* $f_{\mathbf{b}}$ (Belinschi et al., 2017, Theorem 2.2). (In our implementation, we compute $\omega(\mathbf{b})$ by iterating $\omega_i = \mathbf{f}_{\mathbf{b}}(\omega_{i-1})$ until the maximum elementwise difference between the iterates $\omega_i$ does not exceed a specified tolerance parameter $\sim 10^{-6}$.)

Thus, the operator-valued Cauchy transform of $\mathbf{L}_p(w_1, w_2) = \mathbf{b}_0 \otimes 1 + \mathbf{b}_1 \otimes w_1 + \mathbf{b}_2 \otimes w_2$ can be computed by $\mathbf{G}_{\mathbf{L}_p}(\mathbf{b}) = \mathbf{G}_{\mathbf{b}_1 \otimes w_1 + \mathbf{b}_2 \otimes w_2}(\mathbf{b} - \mathbf{b}_0) = G_{\mathbf{b}_1 \otimes w_1}(\omega(\mathbf{b} - \mathbf{b}_0))$.

(4) Finally, the *scalar-valued Cauchy transform $G_p(z)$ of $p(w_1, w_2)$ can be extracted from the first entry of the operator-valued Cauchy transform* $\mathbf{G}_{L_p}$ *of* $\mathbf{L}_p(w_1, w_2)$, *evaluated at a diagonal matrix* $\Lambda_\epsilon(z)$, *as* $\epsilon \downarrow 0$ (Belinschi et al., 2017, Corollary 3.6):

$$G_p(z) = \lim_{\epsilon \downarrow 0}[\mathbf{G}_{L_p}(\Lambda_\epsilon(z))]_{1,1}, \quad \text{where} \quad \Lambda_\epsilon(z) := \begin{pmatrix} z & & & \\ & i\epsilon & & \\ & & \ddots & \\ & & & i\epsilon \end{pmatrix}.$$

(In our implementation, we found that evaluating $\mathbf{G}_{L_p}(\Lambda_\epsilon(z))$ with $\epsilon \sim 10^{-6}$ worked well.)

Thus, the algorithm above allows us to compute the Cauchy transform $G_p$, which completely determines the distribution of $p(w_1, w_2)$. For example, using $G_p$, we can compute the density $f_p$ of $p(w_1, w_2)$ at $x \in \mathbb{R}$ by

$$f_p(x) = \lim_{\epsilon \downarrow 0} \frac{-1}{\pi}\mathrm{Im}(G_p(x + i\epsilon)).$$

For example, see Couillet and Liao (2022, Theorem 2.1). Furthermore, we can compute the point mass $g_p(x)$ at $x \in \mathbb{R}$ (if there is one) by

$$g_p(x) = \lim_{\epsilon \downarrow 0} i\epsilon G_p(x + i\epsilon).$$

## Appendix D. Additional numerical experiments

### D.1. Full-batch diverges, mini-batch converges

Consider the dynamics of full-batch gradient descent with step size $\alpha$, and two-batch gradient descent ($B = 2$) with step size $\alpha/2$. Then it is possible for the full-batch iterate to diverge (i.e. $\|\mathbf{I} - \alpha\frac{1}{n}\mathbf{X}^\mathsf{T}\mathbf{X}\| > 1$), while the two-batch iterate still converges (i.e. $\|\mathbf{I} - \alpha\mathbf{Z}\| > 1$). This is demonstrated by the simple numerical demonstration in the figure on the right, where the entries of $\mathbf{X} \in \mathbb{R}^{n \times p}$ with $n = 1,000$ and $p = 1,500$, noise $\boldsymbol{\eta}$, and $\boldsymbol{\beta}_*$ are (a fixed realization of) i.i.d. standard Gaussians, and a step size of $\alpha = 0.5$ is used.[10]

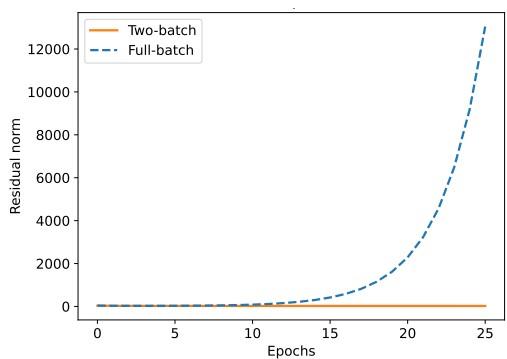

### D.2. Overparameterized regime

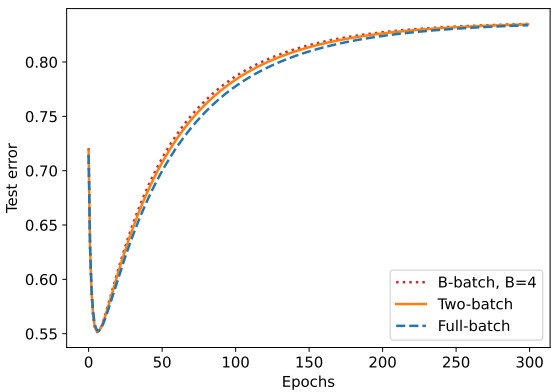
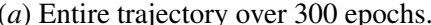

(a) Entire trajectory over 300 epochs.

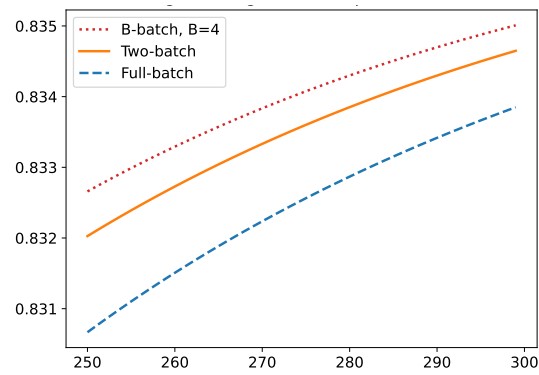

(b) Limiting trajectory in the last 50 epochs.

Figure 2: Empirical generalization error dynamics with standard Gaussian data $\mathbf{X} \in \mathbb{R}^{1,000 \times 1,500}$ ($\gamma = 3/2$), $\sigma = 0.5$, and $\boldsymbol{\beta}_*$ sampled uniformly at random from the unit sphere. Gradient descent with step size $\alpha = 0.2$ compared to $B$-batch gradient descent with step size $\alpha/B$ for $B = 2, 4$. The test error is averaged over $1,000$ simulations with $1,000$ test samples in each.

---

10. This choice of $\alpha$ is slightly larger than $2/(1 + \sqrt{p/n})^2$, which is the almost sure limit of $\|\mathbf{X}\|$ by the Bai-Yin law.

Figure 2 shows the generalization error dynamics of full-batch gradient descent with step size $\alpha$ and $B$-batch gradient descent with step size $\alpha/B$ for $B = 2, 4$ in the overparameterized regime. Overall, the difference is slight (according to the scaling of the figures), highlighting how the full-batch and mini-batch dynamics are matched using the linear scaling rule. Nonetheless, the difference is visually apparent during the middle of training.

For an extreme illustration of the differences that can be caused by large step sizes, recall that we discussed in Remark 6 (and demonstrate in Appendix D.1) that with a larger choice of $\alpha$, full-batch gradient descent can diverge, but two-batch gradient descent still converges.

### D.3. Underparameterized regime

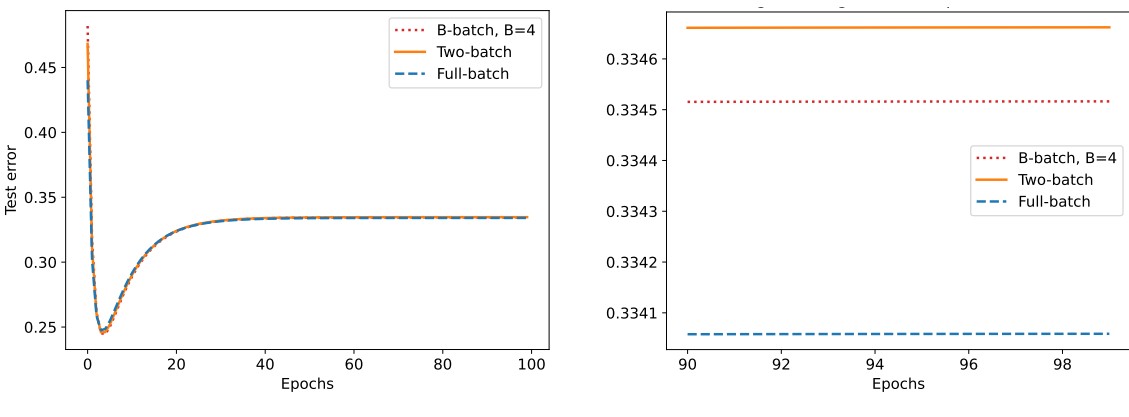

($a$) Entire trajectory over 100 epochs.    ($b$) Limiting trajectory in the last 10 epochs.

Figure 3: Empirical generalization error dynamics with standard Gaussian data $\mathbf{X} \in \mathbb{R}^{4,000 \times 1,000}$ ($\gamma = 1/4$), $\sigma = 1$, and $\boldsymbol{\beta}_*$ sampled uniformly at random from the unit sphere. Gradient descent with step size $\alpha = 0.4$ compared to $B$-batch gradient descent with step size $\alpha/B$ for $B = 2, 4$. The test error is averaged over $1,000$ simulations with $1,000$ test samples in each.

In Figure 3, we compare full-batch gradient descent and mini-batch gradient descent with $B = 2, 4$ mini-batches using the linear scaling rule for the step size in the underparameterized regime. Similar to the observations for Figure 2, the generalization error trajectories of mini-batch and full-batch gradient descent are closely matched. However there are very slight differences; for instance, the limiting risk of two-batch gradient descent is greater by about $\sim 0.05$.

