# OpenReview forum: "Error dynamics of mini-batch gradient descent with random reshuffling for least squares regression"
_algorithmiclearningtheory.org/ALT/2025/Conference — ALT 2025_

### Official Review · Reviewer_TVp9 · 2024-11-02

**Rating:** 4
**Confidence:** 4

**Review:**

This paper provides exact, closed-form analysis for the error dynamics of mini-batch SGD in the context of least squares regression, in the challenging multi-epoch scenario with random reshuffling. The authors’ approach is able to provide analytical expressions for both training and generalization errors, which are generally challenging to derive and hence are notable technical contributions. In particular, this work provides population risk results under random reshuffling, a non-trivial setting that adds complexity to the analysis.

That said, the paper lacks a clear high-level message and main takeaways. The results are heavily technical, primarily consisting of closed-form formulas for error dynamics and population risk, but it is unclear what we fundamentally learn from these formulas. While the authors attempt to offer interpretations in various remarks (e.g., Remarks 4, 5, 6 and corresponding appendices), these interpretations remain rather technical and do not effectively address key aspects such as convergence rates or any generalizable properties of mini-batch SGD without replacement.

Moreover, the analysis is restricted to least-squares linear regression, without discussing the potential extension or relevance of the findings to broader settings, such as convex and/or smooth optimization. This limited scope further weakens the significance of the results, as it remains uncertain whether the conclusions drawn here hold in (even slightly) more general settings.

The most critical issue, however, is the lack of discussion and comparison to prior foundational work in the stochastic optimization literature. The paper mostly cites sources from statistics and deep learning but omits directly relevant papers in optimization, which significantly undermines the significance of its contributions. These include, for example:

1. “Why random reshuffling beats stochastic gradient descent” by Gurbuzbalaban, Ozdaglar, and Parrilo (2015)
2. “Without-replacement sampling for stochastic gradient methods” by Shamir (NeurIPS 2016)
3. “Random shuffling beats SGD after finite epochs” by HaoChen and Sra (ICML 2019)
4. “SGD without Replacement: Sharper Rates for General Smooth Convex Functions” by Nagaraj, Netrapalli, and Jain (ICML 2019)
5. “Closing the convergence gap of SGD without replacement” by Rajput, Gupta, and Papailiopoulos (ICML 2020)

Proper citation and comparison with these works (that apply more generally to convex, smooth objectives rather than only to least-squares regression) are essential for positioning the contribution of this paper and to appreciate its significance.

Overall, while the technical results in this paper are solid and seemingly sound, they lack a compelling overarching narrative, are limited in scope, and most crucially---omit essential context within relevant prior work that is necessary for a comprehensive understanding of their significance. I recommend that the authors refine the framing to emphasize the implications of their results and incorporate a more thorough discussion of related literature, before this work is considered for acceptance.

**Paper Award:**

No

---

> ### Author Response · Authors · 2024-11-20
>
> We thank the reviewer for their insightful comments and suggestions. We will provide more specific responses below.
>
> > That said, the paper lacks a clear high-level message and main takeaways. The results are heavily technical, primarily consisting of closed-form formulas for error dynamics and population risk, but it is unclear what we fundamentally learn from these formulas. While the authors attempt to offer interpretations in various remarks (e.g., Remarks 4, 5, 6 and corresponding appendices), these interpretations remain rather technical and do not effectively address key aspects such as convergence rates or any generalizable properties of mini-batch SGD without replacement.
>
> An important motivating line of work for this paper involves the study of the generalization error of gradient descent/flow, for example [Advani et al., 2020]. The main conclusion are essentially drawn from the idea that the error dynamics (for least squares linear regression) are completely determined by the eigenvalue spectrum of the sample covariance matrix $W = \frac{1}{n} X^T X$ of the features $X$ — e.g., from analogous closed-form expressions as Theorem 2 and Theorem 8, plus more specific assumptions on the model (e.g. $X$ is a Gaussian random matrix), insights into phenomena such as early stopping, implicit regularization, benefits of overparameterization, etc. are drawn. The aim of the paper is to connect the dynamics of mini-batch gradient descent with random reshuffling to full-batch gradient descent, where much more is understood. This is based on connections such as the linear scaling rule (for the learning rate and batch size) to match the dynamics, and an analogous description for the dynamics in terms of a modified cross-covariance matrix $Z$.
>
> We agree that the technical description in terms of $Z$ is not immediately transparent and interpretable, and our aim was to provide more insight into these technical formulas through the various remarks and examples. For example, we consider the asymptotic setting in Section 3.3 to show that mini-batching with random reshuffling effectively results in a multiplicative shrinkage effect on the spectrum of $W$. Therefore, the dynamics, in analogy with the much better understood gradient descent/flow, are comparable to the dynamics of full-batch gradient descent (or sampling with replacement), except that the difference in eigenvalues effectively result in different trajectories (e.g., the early stopped risk at a specific stopping time may be lower for mini-batching with random reshuffling compared to the analogous gradient descent). The shrinkage effect also suggests that the learning rate can be taken to be slightly larger than suggested as the linear scaling rule if smaller batches are used.
>
> Madhu S. Advani, Andrew M. Saxe, and Haim Sompolinsky. High-dimensional dynamics of generalization error in neural networks. Neural Networks, 132:428–446, 2020.
>
> > Moreover, the analysis is restricted to least-squares linear regression, without discussing the potential extension or relevance of the findings to broader settings, such as convex and/or smooth optimization. This limited scope further weakens the significance of the results, as it remains uncertain whether the conclusions drawn here hold in (even slightly) more general settings.
>
> We acknowledge that linear models are limited, but we still believe that their study provides interesting insights that are often observable and sometimes generalizable to more complicated settings (e.g., some references are given in the section on related works on linear models in p.3, including neural networks in a kernel regime, and Reviewer nCo7 has provided additional references to concurrent works on more complicated models such as two-layer networks). The techniques may be applicable to more realistic settings, such as one-layer networks with non-linearities (where linear models provide a basis for many foundational results).
>
> We have not considered the extension to other settings such as convex or smooth optimization; but it appears that the underlying idea of comparing the dynamics of mini-batch gradient descent with random reshuffling to the dynamics of sampling with replacement (or full-batch), and quantifying the difference, should be generalizable. (Indeed, it seems that this idea is used in some of the provided references in the stochastic optimization literature.)

---

> > ### Author Response · Authors · 2024-11-20
> >
> > > The most critical issue, however, is the lack of discussion and comparison to prior foundational work in the stochastic optimization literature. The paper mostly cites sources from statistics and deep learning but omits directly relevant papers in optimization, which significantly undermines the significance of its contributions.
> >
> > We agree that a better discussion of key works on stochastic gradient descent with random reshuffling from the stochastic optimization literature is fitting in the related works (on sampling without replacement). We thank the reviewer for their additional references in this direction, and we will expand our discussion with the help of these suggestions.
> >
> > We will try to provide a brief explanation of some differences in our results and those typically obtained in the stochastic optimization literature. The motivation for this paper was on connecting the dynamics of mini-batch gradient descent with random reshuffling with the much better understood dynamics of full-batch gradient descent/flow (e.g., as studied in the cited literature from statistics/machine learning theory). As a result, the focus is different to some of the works in the stochastic optimization literature: for example, we have aimed to provide an exact description of the dynamics in terms of the spectrum, which has been important for understanding the effects of early stopping, implicit regularization, benefits of overparameterization, etc. for gradient descent/flow (e.g., by analyzing the formulas under a class of random matrix models). On the other hand, we have not focused on bounding the complexity/convergence rate of the specific optimization problem. However, it would be interesting extension to see how our results intersect with the results on random reshuffling from the stochastic optimization literature (e.g., as a particular example of a quadratic/strongly convex objective, which is considered as a special case in some of the cited works).

---

### Official Review · Reviewer_RZLR · 2024-11-06

**Rating:** 6
**Confidence:** 3

**Review:**

**Summary**

The paper studies SGD under the random reshuffling scheme for the linear regression problem. Concretely, the authors provide the limit of the iterate (average in every epoch) and study the generalization error. In addition, the authors also give some asymptotic analysis on the batching. In summary, I think the problem is worth studying since random reshuffling is commonly implemented in practice but with fewer theoretical studies.

**Major points**

In Eq. (B.2), should $\prod_{j:\tau(j)<\tau(b)}(I-\alpha W_{\tau (j)})X_b^\top \eta_b$ be $\prod_{j:j>\tau^{-1}(b)}(I-\alpha W_{\tau (j)})X_b^\top \eta_b$ or $\prod_{j:j>b}(I-\alpha W_{\tau (j)})X_{\tau(b)}^\top \eta_{\tau(b)}$? I can't see why the current form holds. Could the authors elaborate more on this equation? Even under the current form, the following derivation seems incorrect because the current $W_{\tau(j)}$ is also different from $W_{j}$ in Eq. (3.1).

Since the above issue is a fundamental step in the paper, I can't verify the subsequent proof in a further step. I would be happy to improve my score if the authors could provide a convincing explanation during the feedback period.

**Minor points**

1. Page 21, the third line in the proof of Theorem 2, the second $\beta_k^{(b)}$ should be $\beta_k^{(b-1)}$.

2. Though the paper focuses on the linear model with random reshuffling, many works considering the general case are missing, which I believe should be also added as the background. Here, I list some of them but not all. The authors should be more careful

    - Gurbuzbalaban, Mert, Asu Ozdaglar, and Pablo A. Parrilo. "Convergence rate of incremental gradient and incremental Newton methods." SIAM Journal on Optimization 29.4 (2019): 2542-2565.

    - Haochen, Jeff, and Suvrit Sra. "Random shuffling beats SGD after finite epochs." International Conference on Machine Learning. PMLR, 2019.

    - Nguyen, Lam M., et al. "A unified convergence analysis for shuffling-type gradient methods." Journal of Machine Learning Research 22.207 (2021): 1-44.

    - Nagaraj, Dheeraj, Prateek Jain, and Praneeth Netrapalli. "Sgd without replacement: Sharper rates for general smooth convex functions." International Conference on Machine Learning. PMLR, 2019.

**Paper Award:**

No

---

> ### Author Response · Authors · 2024-11-18
>
> Thank you for your helpful comments.
>
> **Major point**
>
> *(This response to the major point is incorrect, and the point raised by the reviewer is correct. Please refer to the later response dated 20 Nov 2024 for the correct explanation.)*
>
> To address the major point, Eq. (B.2) should read $\prod_{j: \tau(j) < \tau(b)} (I - \alpha W_j) X_b^T \eta_b$; that is, $W_{\tau(j)}$ should read $W_j$ in this product. This change should also be made for similar terms in the equations that appear after this (in this proof only, namely equations (B.3), (B.4), (B.5) and (B.7)) to align everything with the definition of $\Pi_b$ in Eq. (3.1), which we anchor as the starting point (based on our choice of notation). The rest of the proof remains otherwise unchanged.
>
> To explain our notation (with the correction above) in the proof, it might help if we provide a simple example with $B = 3$. As a reminder of our convention (from footnote 2 on page 5), we identify each permutation $\tau \in S_B$ with a list $(\tau(1), \tau(2), \ldots, \tau(B))$ of matrices that are multiplied from right to left in the product, and take the product over an empty set to be the identity.
> * Consider the permutation $\tau = (3, 2, 1)$ – i.e., $\tau(1) = 3$, $\tau(2) = 2$, and $\tau(3) = 1$ [fixed] – this means that with this ordering, the third batch appears first, then the second batch, then the first batch.
> * The noise terms associated with each mini-batch are of the form $X_b^T \eta_b$. Thus, the final "noise terms" that appear from this process at the end of this epoch will be $X_3^T \eta_3$, $(I - \alpha W_3) X_2^T \eta_2$, and $(I - \alpha W_3) (I - \alpha W_2) X_1^T \eta_1$ respectively. The prefactors of the form $(I - \alpha W_j)$ for a specific mini-batch $X_b^T \eta_b$ correspond to the mini-batch $j$ that appear before $b$ in the permutation, i.e., the mini-batch $j$ such that $\tau(j) < \tau(b)$.
> * In our notation, the sum of the "noise terms" above can be written $\sum_{b=1}^B \prod_{j: \tau(j) < \tau(b)} (I - \alpha W_j) X_b^T \eta_b$. This explains why it should look like a product of terms of the form $(I - \alpha W_j)$, and not $(I - \alpha W_{\tau(j)})$.
>
> Please do let us know if we can provide further explanation.
>
> **Minor points**
> 1. Thank you for noticing this typo.
> 2. Thank you for the additional references on works that have also studied random reshuffling. We will look through these and provide a more extensive coverage of these related works.

---

> > ### Comment · Reviewer_RZLR · 2024-11-18
> > **Response**
> >
> > I thank the authors for the quick feedback. However, I still have questions.
> >
> > 1. I think $\tau(3)=3$ in your comment should be $\tau(3)=1$?
> >
> > 2. If my point in 1. above is correct, it seems the terms should be $X_1^\top \eta_1$, $(I-\alpha W_1)X_2^\top\eta_2$, $(I-\alpha W_1)(I-\alpha W_2)X_3^\top\eta_3$. However, $\sum_{b=1}^B\prod\_{j: \tau(j)<\tau(b)} (I-\alpha W_j)X_b^\top \eta_b=(I-\alpha W_3) (I-\alpha W_2)X_1^\top \eta_1+ (I-\alpha W_3)X_2^\top \eta_2+X_3^\top \eta_3$.
> >
> > In addition, I found two typos in my review, the subscript $j<...$ should instead be $j>...$, I apologize if it leads to any confusion and have fixed them.

---

> ### Author Response · Authors · 2024-11-18
>
> *(This response is incorrect. Please refer to the later response dated 20 Nov 2024 for the correct explanation.)*
>
> My apologies, it should be $\tau(3) = 1$ and I have fixed this to avoid confusion.
>
> The idea used in the proof is that a permutation $\tau = (3, 2, 1)$ gives the order of the mini-batches read from left to right. The notation should be consistent with this notion throughout, and I apologise for the typographic errors in making this precise.
>
> By directly iterating the first displayed equation $\beta_k^{(b)} = \beta_k^{(b-1)} - \frac{B \alpha}{n} X_{\tau(b)}^T (X_{\tau(b)} \beta_k^{(b-1)} - y_{\tau(b)})$ – i.e., the mini-batch $\tau(1)$ is used first and then $\tau(2)$ and then $\tau(3)$ and so on – the term that appears at the end of Eq. (B.2) should be $$\sum_{b=1}^B \prod_{j: j < b} (I - \alpha W_{\tau(j)}) X_{\tau(b)}^T \eta_{\tau(b)}. \quad (I)$$
> That is, continuing with our running example $B = 3$ and $\tau = (3, 2, 1)$, we should get $X_3^T \eta_3 + (I - \alpha W_3) X_2^T \eta_2 + (I - \alpha W_3) (I - \alpha W_2) X_1^T \eta_1$.
> Now we want to rewrite this so that the indexing is consistent with summands of the form $X_b^T \eta_b$. The correct notation, consistent with how $\tau$ determines the ordering, is
> $$\sum_{b=1}^B \prod_{j: j < \tau^{-1}(b)} (I - \alpha W_{\tau(j)}) X_{b}^T \eta_{b}. \quad (II)$$
> That is, Equations (I) and (II) here are identical, and (II) should appear as Eq. (B.2). (An additional line explaining this change of indexing in the proof should make this step much clearer and easier to parse.)
>
> This calculation shows that the intended indexing should be $j: j < \tau^{-1}(b)$: in words, $\tau^{-1}(b)$ gives the correct position of mini-batch $b$ in the ordering $\tau$, and $j < \tau^{-1}(b)$ selects the mini-batches $W_{\tau(j)}$ that appear before it. (Other indexing is definitely possible; this way is consistent with the fact that the matrices are also multiplied in the right order, i.e., $\tau(1)$ then $\tau(2)$ then $\tau(3)$.)
>
> To summarize: the correct indexing consistent with the notion that $\tau$ determines the order when read from left to right should be $j: j < \tau^{-1}(b)$. The rest of the proof is unaffected by these notational corrections since it is based on this idea. (The notation in Eq. (3.1) should also be fixed so $\Pi_b$ is defined by $\Pi_b = \frac{1}{B!} \sum_{\tau \in S_B} \prod_{j: j < \tau^{-1}(b)} (I - \alpha W_{\tau(j)})$.)
>
> Thank you for pointing out the typos in the notation for this notion. I hope this clears up the intention and that the approach can be understood now.

---

> > ### Comment · Reviewer_RZLR · 2024-11-18
> > **Response**
> >
> > I thank the authors' quick response. My comments are as follows:
> >
> > - First, I want to remark that the notation/equation in the authors' current response is identical to my initially submitted review.
> >
> > - Next, I kindly remind the authors that this notation/equation still has mistakes as mentioned, which is the subscript $j<...$ should be $j>...$ instead. I have pointed out this mistake by myself in the prior response. However, the authors repeat the same mistake here again, so I hope the authors could be more careful.
> >
> > - Moreover, I respectfully disagree with the authors calling this issue ''typos''. In contrast, I think this is an important problem since it affects the whole paper and leads to critical confusion (at least to me). This is also why I list it as a major point.
> >
> > In summary, the basic point in my initially submitted review still holds, i.e., the notation has a big mistake. I believe the whole paper needs a major revision to fix this issue. As such, I prefer to maintain my current score but will pay close attention to any possible discussion between the authors and other reviewers to decide my final rate.

---

> > > ### Author Response · Authors · 2024-11-20
> > >
> > > I apologize for the delay in response; since my reply before has errors as kindly pointed out by the reviewer, I have carefully checked the details before responding.
> > >
> > > Indeed, I stated a wrong expression for the running example I was describing: for $\tau = (3, 2, 1)$, the correct terms should have been $X_1^T \eta_1$, $(I - \alpha W_1) X_2^T \eta_2$, and $(I - \alpha W_1) (I - \alpha W_2) X_3^T \eta_3$ as the reviewer had correctly pointed out. This aligns with the correction that in Eq. (B.2), the indexing in the second sum should be correctly written as $\sum_{j: j > \tau^{-1}(b)} (I - \alpha W_{\tau(j)}) X_b^T \eta_b$, as the reviewer had correctly identified initially.
> > >
> > > Again, I want to apologize to the reviewer for this careless mistake (which led to the impression that the fix was purely notational), and would like to thank them again for their attentiveness to identify this issue in the proof.
> > >
> > > Having carefully checked the details, I would just like to explain that the proof of Theorem 2, which relies on Eq. (B.2), holds essentially as written after correcting the indexing in the equations from $j < …$ to $j > …$ (specifically, in Eqs. (B.2), (B.3), (B.4), (B.5), (B.7)).
> > > * The main difficulty that could arise with this correction is that the definition of $\Pi_b$ in (3.1), which uses this incorrect indexing, could be problematic. However, the same definition of $\Pi_b$ (using the indexing $j: j < \tau^{-1}(b)$) can be used in the corrected proof: that is, we have the equivalences
> > > $$\Pi_b = \frac{1}{B!} \sum_{\tau \in S_B} \prod_{j: j < \tau^{-1}(b)} (I - \alpha W_{\tau(j)}) = \frac{1}{B!} \sum_{\tau \in S_B} \prod_{j: j > \tau^{-1}(b)} (I - \alpha W_{\tau(j)}).$$
> > > This is because $\Pi_b$ is defined as a sum over the symmetric group, so every permutation $\tau$ in the first sum can be matched in a one-to-one correspondence to a permutation $\tau’$ in the second sum (e.g. if $\tau = (1, \dots, b-1, b, b+1, \dots, B)$, then $\tau’ = (b+1, \dots, B, b, 1, \dots, b-1)$ gives the same summand).
> > > * A second change that is not just notation is in the proof of the claim (B.7): in fact, the proof of the claim with $j: j > …$ on the left hand side is actually much simpler; the same argument used for the fixed permutation $\tau$ works, and the reversal $\tau’$ is actually not needed (the reversal is to do with the incorrect order that arises from the incorrect notation).
> > >
> > > To summarize the technical details above, I hope that I have explained that this issue in the proof of Theorem 2 is local in nature because it can be fixed by correcting the indexing (and clarifying the exposition). The rest of the paper is unaffected after correcting this issue.

---

> > ### Comment · Reviewer_RZLR · 2024-11-21
> > **Response**
> >
> > I thank the authors for their response. However, I think some points still need to be discussed further. For simplicity, I use the old notation to refer to the original notation provided in the paper, the new1 notation to refer to the notation under $j<...$, and the new2 notation to refer to the notation under $j>...$.
> >
> > I want to check whether my understanding of the authors' response is correct:
> >
> > 1. The authors first intended to say that the proof of Theorem 2 (here I mean starting from Eq. (B3)) still holds even under changing the old notation to the new1 notation.
> >
> > 2. The second point by the authors is that If we only focus on $\Pi_b$, then the definition of it under the new1 and new2 notations are the same (I like the author's one-to-one mapping argument since it's elegant). So if the proof under the new1 notations goes through, so does it under the new2 notations.
> >
> > If the above two points hold, however, I am kind of confused now because I didn't see any reasoning (even informal) from the authors' previous responses for the first point. The only thing I can find is a simple example saying the old and new1 notations are aligned for $\Pi_b$ in that case. Did I miss anything? Moreover, if the proof holds under the new1 notation, why did the authors mention the second point in their response (i.e., about Eq. (B.7))? Based on my understanding, this argument seems residual once the authors claimed the proof goes well under the new1 notation and the new1 and new2 notations are equivalent for $\Pi_b$. Would the authors explain more about my confusion?

---

> ### Author Response · Authors · 2024-11-22
>
> We thank the reviewer for their quick response.
>
> Our last response is meant to convey that
> 1. The proof of Theorem 2, starting from Eq. (B.2), is exactly the same if the old notation is changed to the correct new2 notation pointed out by the reviewer. (The only other change is to the proof of the claim Eq. (B.7) explained, and expanded on below.)
> 2. The definition of $\Pi_b$ is the same under the new1 or new2 notations.
>
> The previous (incorrect) responses attempted to explain (a variant of) point 1, saying that the proof holds if the old notation was changed to the new1 notation. This was not quite right, as the reviewer pointed out, since the notation with new1 is inconsistent with the actual order of the iterations; i.e., in deriving Eq (B.2) from the displayed equations beforehand, leading to a flipped ordering that is carried forward. I apologize for this careless mistake, and I hope that our last response provides a careful answer to the reviewer's question. (I've made a brief edit at the previous responses to highlight the error.)
>
> The point about Eq. (B.7) was to explain all the other changes to the proof (besides changing the notation to new2) to highlight that it is not a global error. To say in words, it says that with the correct new2 notation, the proof of the claim Eq. (B.7) following the displayed equation only needs to be modified by removing the reversing argument for $\tau'$ (this is essentially the swapping trick).

---

> > ### Comment · Reviewer_RZLR · 2024-11-22
> > **Response**
> >
> > Thanks for the further clarification. Now, I understand the authors' points and have decided to raise my score. However, I don't know why I cannot edit my review (as such I cannot say the final grade since I need to see the text after each score). I will check OpenReview later. If the system still cannot work, I will write a direct message to the AC.
> >
> > Another minor comment: the reason that the proof still works under the new2 notation (actually even the new1 notation) is the equivalence of $\Pi_b$ under these three notations as far as I can check, so maybe the simplest way to tackle this notations issue is only to change Eq. (B2) and prove the equivalence in Appendix B.1. But this is only my personal view, how to change the writing is up to the authors.

---

> > > ### Author Response · Authors · 2024-11-22
> > >
> > > Thank you for your response and suggestion. Your suggestion is very reasonable: the proof of the result is written so that it is most natural with the new2 notation (but it could really be set up in a different order). We thank the reviewer again for their attentiveness in pointing out this error so that it can be corrected and so the proof does not lead to confusion.

---

### Official Review · Reviewer_nCo7 · 2024-11-11
**Solid contribution**

**Rating:** 7
**Confidence:** 3

**Review:**

This work investigates the dynamics of mini-batch gradient descent with random reshuffling for well-specified least squares regression. The main contribution is to identify a sample cross-covariance matrix which quantifies the impact of random reshuffling in the convergence and the error dynamics compared to full-batch and other replacement-based schemes. The paper discusses different limits where this sample cross-covariance matrix - given by a cumbersome non-commutative polynomial of the batches sample-covariance matrices - simplifies, such as small step-sizes, large sample size and the proportional regime. The main conclusion is that sampling without replacement can have a significant impact in the covergence and generalization error in the regime of constant step-size.

Overall, this is a solid theoretical paper with, to the best of my knowledge, novel results for SGD without replacement. The manuscript is well written, though dense and technical. Of course, the setting (well-specified least-squares) is limited in scope, but it is a fair start given the limited literature in this direction.

**Questions**:
- One of the interesting points in the paper is the explicit characterization of the learning rate dependence in the trajectory and in particular in the limit. I miss, however, a discussion on the implications of this dependence in terms of the implicit bias of the solution. For instance, how does the performance depends on the learning rate? Is it the dependence monotonic? Is there any benefit of tuning the learning rate in this setting? Is it comparable to adding a penalty, for instance?

- In the introduction:

> *Most prior theoretical works also analyze gradient descent with infinitesimal learning rates, i.e. gradient flow (Advani et al., 2020; Ali et al., 2020). In this work, we analyze the dynamics of gradient descent with arbitrary learning rates.*

In this level of generality, this sentence is misleading. There is a very extensive body of literature about SGD with constant learning rate, starting as early as 1986 [1] to nowadays, see e.g. [2-7] for a few. These works span different settings (one-pass, two-layer neural networks, etc.), but are closely related to the discussion here.

- On the related works, page 4:

> "From a dynamical perspective, Paquette et al. (2021); Lee et al. (2022); Paquette et al. (2022) show that the trajectories of SGD for ridge regression with finite step sizes and high-dimensional random data concentrate on a deterministic function determined by a Volterra equation, assuming the batch sizes are vanishingly small as a fraction of the sample size."

These works show that low-dimensional functionals of the SGD trajectories (such as the risk) concentrate to a deterministic function solving a Volterra equation. Alternatively, this can be seen as convergence of the trajectory (in a weak sense) to an effective stochastic process in the high-dimensional limit (e.g. Theorem 6 in Paquette et al. (2022)). It is worth mentioning that the results in these works for streaming SGD are concurrent to [2, 4, 6, 7], who derived equivalent concentration results in the constant step-size, high-dimensional regime a larger class of problems (such as two-layer neural networks) that include least-squares as a particular case.

- Related to the above, one of the observations in the line of work [2, 4, 6,7] is that for streaming SGD, taking a constant step size in the high-dimensional regime can change the fixed point of the dynamics, usually for the worst (e.g. by estabilizing an unstable fixed point or shifting the global minimum). In the least-squares case, where there is only a single global minimum of the population risk, this is just a manifestation that the process converges to a stationary distribution with finite variance (hence the excess population risk is bounded away from zero). How does this compare with the random reshuffling case studied here? In particular, from Corollary 9, are there cases where a finite learning rate leads to a better performance than zero learning rate?


**References**:
- [1] Georg Ch. Pflug. Stochastic minimization with constant step-size: asymptotic laws. SIAM Journal on Control and Optimization, 24(4):655–666, 1986.

- [2] David Saad, Sara Solla. Dynamics of On-Line Gradient Descent Learning for Multilayer Neural Networks. Part of Advances in Neural Information Processing Systems 8 (NIPS 1995)

- [3] Dieuleveut, Aymeric, and Francis Bach. Nonparametric stochastic approximation with large step-sizes. Ann. Statist. 44(4): 1363-1399 (August 2016).

- [4] Sebastian Goldt, Madhu Advani, Andrew M. Saxe, Florent Krzakala, Lenka Zdeborová. Dynamics of stochastic gradient descent for two-layer neural networks in the teacher-student setup.  Part of Advances in Neural Information Processing Systems 32 (NeurIPS 2019).

- [5] Aymeric Dieuleveut, Alain Durmus and Francis Bach. Bridging the gap between constant step size stochastic gradient descent and Markov chains. The Annals of Statistics, 48 (3): 1348 – 1382, June 2020.

- [6] Gerard Ben Arous, Reza Gheissari, Aukosh Jagannath. High-dimensional limit theorems for SGD: Effective dynamics and critical scaling.  Part of Advances in Neural Information Processing Systems 35 (NeurIPS 2022).

- [7] Luca Arnaboldi, Ludovic Stephan, Florent Krzakala, Bruno Loureiro. From high-dimensional & mean-field dynamics to dimensionless ODEs: A unifying approach to SGD in two-layers networks. Proceedings of Thirty Sixth Conference on Learning Theory, PMLR 195:1199-1227, 2023.

**Paper Award:**

No

---

> ### Author Response · Authors · 2024-11-20
>
> We thank the reviewer for the detailed and insightful review. We will try and provide answers to the questions raised:
>
> > One of the interesting points in the paper is the explicit characterization of the learning rate dependence in the trajectory and in particular in the limit. I miss, however, a discussion on the implications of this dependence in terms of the implicit bias of the solution. For instance, how does the performance depends on the learning rate? Is it the dependence monotonic? Is there any benefit of tuning the learning rate in this setting? Is it comparable to adding a penalty, for instance?
>
> The dependence on the learning rate $\alpha$ is implicitly embedded in the matrix $Z$ (say, in Theorem 2 and Theorem 8 ), so that it differs from the usual sample covariance matrix $W$ that determines the trajectory for gradient descent/flow. In general, we don’t have a precise description of this implicit bias (other than noting that the difference is of order $O(\alpha)$, keeping all else fixed). We do show that it effectively results in a (multiplicative) shrinkage effect on the spectrum of $W$ under more assumptions (i.e., asymptotically). This differs from penalties such as ridge, where there is an additive effect on the spectrum. This provides more quantitative insights into the effect on the trajectory since the trajectory of gradient descent for a linear model is essentially determined by the spectrum of $W$.
>
> In terms of monotonicity, one could impose more assumptions and get explicit formulas (e.g., Proposition 12 in the underparameterized regime) that show how the error depends on the learning rate and spectrum of $W$. So, one could deduce that at an epoch where mini-batching has lower risk than the full-batch, increasing the learning rate would increase the difference.
> In terms of tuning the learning rate, the analysis also highlights the importance of the linear scaling rule for the learning rate as a function of the batch size, and the shrinkage effect suggests that a slightly larger learning rate can be used with smaller batches.
>
> > In this level of generality, this sentence is misleading. There is a very extensive body of literature about SGD with constant learning rate, starting as early as 1986 [1] to nowadays, see e.g. [2-7] for a few. These works span different settings (one-pass, two-layer neural networks, etc.), but are closely related to the discussion here
>
> We will clarify this statement so that it conveys our approach, and not its originality.
>
> > These works show that low-dimensional functionals of the SGD trajectories (such as the risk) concentrate to a deterministic function solving a Volterra equation. Alternatively, this can be seen as convergence of the trajectory (in a weak sense) to an effective stochastic process in the high-dimensional limit (e.g. Theorem 6 in Paquette et al. (2022)). It is worth mentioning that the results in these works for streaming SGD are concurrent to [2, 4, 6, 7], who derived equivalent concentration results in the constant step-size, high-dimensional regime a larger class of problems (such as two-layer neural networks) that include least-squares as a particular case.
>
> Thank you for the additional references — these recent concentration results for stochastic dynamics for a wider class of problems are definitely interesting examples of applying the approach to study more complicated models than linear models, and we can provide some pointers to these concurrent works to improve the discussion.

---

> > ### Author Response · Authors · 2024-11-20
> >
> > > Related to the above, one of the observations in the line of work [2, 4, 6,7] is that for streaming SGD, taking a constant step size in the high-dimensional regime can change the fixed point of the dynamics, usually for the worst (e.g. by estabilizing an unstable fixed point or shifting the global minimum). In the least-squares case, where there is only a single global minimum of the population risk, this is just a manifestation that the process converges to a stationary distribution with finite variance (hence the excess population risk is bounded away from zero). How does this compare with the random reshuffling case studied here? In particular, from Corollary 9, are there cases where a finite learning rate leads to a better performance than zero learning rate?
> >
> > In the overparameterized regime, which is typically the more interesting setting in this context, the limits are generically the same, e.g., if $X$ is full rank (p. 7), and so the difference in the limit (Corollary 4 or 9) with finite learning rate and zero learning rate is not so interesting. What is interesting in this regime is that the trajectories are different, meaning that with finite learning rate, the minimum risk over the trajectory (with early stopping), or the risk at a prescribed stopping time, could be lower than the gradient flow case. This is due to the shrinkage effect on the spectrum, which also affects the rate of convergence to the limit. (A picture showing a simple demonstration of this can be found in Figure 2 in Appendix D.2 on p.32.) However, we currently don’t have an analytical understanding of the trajectory as a function of the learning rate and the spectrum for the mini-batch case that would make this precise (e.g., assuming a random matrix model in the proportional regime), and this is an interesting area for future work.
> >
> > In the underparameterized regime, the limits can differ. Some simple examples can be found numerically (e.g., even for Gaussian matrices; for a picture, see Figure 3 in Appendix D.3. on p.33). We have observed that the limiting risk still tends to be higher, which is consistent with the intuition that the dependence in the learning process will amplify the noise that is learnt at the end. However, the minimum risk over the trajectory can be lower. It would also be interesting to have a better analytical understanding here too.

---

> > > ### Comment · Reviewer_nCo7 · 2024-11-26
> > >
> > > I thank the authors for their rebuttal that addressed my comments. I am keeping my positive evaluation.

---

### Meta-Review · Area_Chair_QgmX · 2024-12-13

**Recommendation:** Accept
**Confidence:** 3

**Metareview:**

This paper studies the dynamics of mini-batch gradient descent for the least squares problem with random shuffling.

The main contribution of the paper is an explicit characterization of the dynamics of mean iterate generated by mini-batch gradient descent (averaged over the choice of the random permutation used to shuffle the batches) as well as the generalization error of the mean iterate. Both the dynamics and the generalization error depend on a complicated non-commutative polynomial involving the feature matrix $X$. The authors consider various interesting limits such as the large-sample limit (sample size $n \rightarrow \infty$), the proportional limit (feature dimension and sample size diverge proportionally), and the small step-size limit to simplify their characterization and obtain insights on the distinctions between mini-batch gradient descent with random shuffling and full-batch gradient descent.

Evaluation: This paper received one highly positive score, a borderline positive score, and a very negative score. There wasn't a lot of consensus about this paper among the reviewers. This paper appears to be borderline, and one could make reasonable arguments for accepting or rejecting it. My recommendation is to accept the paper due to the following reasons:

1) All three reviewers appreciated that the paper tackles the challenging and practically important problem of analyzing SGD under random shuffling.

2) A fair criticism of this paper, which was raised by one of the reviewers, is its narrow focus on the least squares problem, raising the question of whether there is any hope of generalizing the findings to more general scenarios. Many readers from the optimization community, who tend to analyze SGD by providing convergence bounds under quite general conditions (such as convex losses), would share this concern. The paper takes a different approach of providing an exact characterization of dynamics (rather than a convergence bound) under a stylized setup. This style of analysis is appreciated in the high-dimensional stats/statistical physics/probability community, who would likely find this paper interesting.

3) Another concern raised by the reviewer is that the authors need to adequately discuss important works on analyzing convergence rates for SGD developed in the stochastic optimization community. I think the authors should absolutely do so, although because of the difference in perspectives highlighted above, I do not anticipate that prior results will subsume their results.

4) Lastly, I think this work has the potential to generate some follow-up work. In particular, this work shows that the generalization error and the dynamics of the mean iterate depend on a complex non-commutative polynomial of the dataset $X$. Under a random design and proportional asymptotic assumption, the authors analyze the behavior of this polynomial using a random matrix in the special case of two batches. The general case appears to be a challenging and interesting random matrix theory problem, which might be of interest to readers.

5) I should also highlight the fact that the paper contained an error that was caught by one of the reviewers. The authors fixed it during the review process, and the reviewer seems convinced of the fix.

A suggestion from the meta-reviewer: it would be to clarify why is it reasonable to study the generalization error of the mean iterate (averaged over random shuffling) rather than the expected generalization error of an iterate produced by a single random shuffling, where the expectation over the shuffling is taken at the end? Are these two quantities expected to be similar?

**Paper Award:**

No